# Neddylation inhibition induces glutamine uptake and metabolism by targeting CRL3$^{SPOP}$ E3 ligase in cancer cells

Qiyin Zhou[1,2,3,4,5], Wenyu Lin[2], Chaoqun Wang[2,6], Fei Sun[7], Siwei Ju[8], Qian Chen[1,4], Yi Wang[7], Yongxia Chen[8], Haomin Li [9], Linbo Wang [8], Zeping Hu [7], Hongchuan Jin [2,5], Xian Wang[2,3,5,11 ✉] & Yi Sun [1,4,5,10,11 ✉]

Abnormal neddylation activation is frequently observed in human cancers and neddylation inhibition has been proposed as a therapy for cancer. Here, we report that MLN4924, a small-molecule inhibitor of neddylation activating enzyme, increases glutamine uptake in breast cancer cells by causing accumulation of glutamine transporter ASCT2/SLC1A5, via inactivation of CRL3-SPOP E3 ligase. We show the E3 ligase SPOP promotes ASCT2 ubiquitylation, whereas SPOP itself is auto-ubiquitylated upon glutamine deprivation. Thus, SPOP and ASCT2 inversely regulate glutamine uptake and metabolism. SPOP knockdown increases ASCT2 levels to promote growth which is rescued by ASCT2 knockdown. Adding ASCT2 inhibitor V-9302 enhances MLN4924 suppression of tumor growth. In human breast cancer specimens, SPOP and ASCT2 levels are inversely correlated, whereas lower SPOP with higher ASCT2 predicts a worse patient survival. Collectively, our study links neddylation to glutamine metabolism via the SPOP-ASCT2 axis and provides a rational drug combination for enhanced cancer therapy.

[1] Cancer Institute, the Second Affiliated Hospital, Zhejiang University School of Medicine, Hangzhou 310009, China. [2] Key Lab of Biotherapy in Zhejiang, Sir Run Run Shaw Hospital, Zhejiang University School of Medicine, Hangzhou 310016, China. [3] Department of Medical Oncology, Sir Run Run Shaw Hospital, Zhejiang University School of Medicine, Hangzhou 310016, China. [4] Institute of Translational Medicine, Zhejiang University School of Medicine, Hangzhou 310029, China. [5] Cancer Center, Zhejiang University, Hangzhou 310058, China. [6] Department of Pathology, Affiliated Dongyang Hospital of Wenzhou Medical University, Dongyang 322100, China. [7] School of Pharmaceutical Sciences, Tsinghua-Peking Joint Center for Life Sciences, Beijing Frontier Research Center for Biological Structure, Tsinghua University, Beijing 100084, China. [8] Department of Surgical Oncology, Sir Run Run Shaw Hospital, Zhejiang University, Hangzhou 310016, China. [9] Children's Hospital, Zhejiang University School of Medicine, National Clinical Research Center for Child Health, Hangzhou 310052, China. [10] Research Center for Life Science and Human Health, Binjiang Institute of Zhejiang University, Hangzhou 310053, China. [11]These authors jointly supervised this work: Xian Wang, Yi Sun. ✉email: wangx118@zju.edu.cn; yisun@zju.edu.cn

The process of cell metabolism is precisely controlled for the maintenance of normal tissue homeostasis[1–3]. Oncogenic transformation and rapid tumorigenic growth impose cancer cells a heavy demand of metabolic reprogramming, which involves the alterations of many key signaling pathways[3–6].

Glutamine, the most abundant amino acid in blood and muscle, and the donor for both carbon and nitrogen, is metabolically converted into α-ketoglutarate (α-KG) to enter tricarboxylic acid (TCA) cycle, providing cells with energy, building blocks, and redox homeostasis[7]. Moreover, glutamine is also a signaling molecule that is associated closely with intracellular proliferating pathways such as mammalian target of rapamycin complex 1 (mTORC1) to promote tumor growth[8]. Therefore, the pleiotropic glutamine is a key nutrient supply for accelerated growth of cancer cells, and for the maintenance of the tumorigenic state[4,9]. Indeed, a high glutamine demand has been frequently seen in many human cancer cells[8,10,11]. Moreover, glutamine addiction exhibited by cancer cells often leads to severe glutamine shortage in the tumor microenvironment[12–15].

Glutamine is imported across the plasma membrane through several members of solute-linked carrier families (SLC)[10,16]. Among these transporters, alanine-serine-cysteine transporter 2 (ASCT2; SLC1A5) has high affinity and specificity for glutamine and acts as a key glutamine transporter[16]. Not surprised, ASCT2 is remarkably upregulated in a variety of cancer types (i.e., carcinomas of breast, lung, liver, etc.) and is positively associated with poor patient survival. Furthermore, inactivation of ASCT2 via genetic knockdown or pharmacological inhibition significantly impedes glutamine uptake and attenuates the proliferation and survival of cancer cells[17,18]. Therefore, ASCT2 is an attractive target for antitumor therapy. However, the signaling pathways that regulate ASCT2-mediated glutamine uptake are not fully understood[19], and it is largely unknown whether and how the stability of ASCT2 is regulated by which E3 ubiquitin ligase under what physiological or pathological conditions. Answers to these questions would have significant implications for effective cancer interventions via targeting glutamine transportation.

Neddylation is a type of post-translational modification that attaches NEDD8, a ubiquitin-like peptide, onto a lysine residue of the substrate protein for the modulation of its activity or function. Neddylation is catalyzed by an E1 NEDD8-activating enzyme (NAE), an E2 NEDD8-conjugating enzyme, and an E3 neddylation ligase[20,21]. These neddylation enzymes, along with NEDD8, are over-expressed in many types of human cancers, which is associated with the poor survival of patients. Thus, neddylation was validated as an attractive anti-cancer target[22,23]. Indeed, MLN4924, also known as pevonedistat, the first-in-class small molecular inhibitor of NAE[24], is currently in a number of phases I/II clinical trials as a single agent or in combination with chemo drugs[22]. Neddylation was also found to regulate various metabolism processes, including adipogenesis[25], fatty acid β-oxidation[26], nucleotide metabolism[27], and cellular redox status[28]. Recently, we found that inactivation of neddylation remarkably alters global metabolic profiling via inhibiting mitochondrial functions and promoting glycolysis[29–31]. However, whether and how neddylation regulate amino acid metabolism remain elusive.

Cullin-RING ligases (CRLs) are the largest family of ubiquitin E3 ligases that regulate many important physiological and pathological processes by targeting cellular proteins for proteasome degradation[32]. A typical CRL E3 consists of a scaffold cullin with 8 family members (cullins 1, 2, 3, 4A, 4B, 5, 7, and 9), an adaptor protein with many members, a substrate-recognizing receptor with many members and a RING with two family members (RBX1 and RBX2/SAG)[32]. Cullin family members are

physiological substrates of neddylation, and cullin neddylation is required for CRL activity[33].

Speckle-type POZ protein (SPOP) is a substrate adaptor for the Cullin 3-based E3 ligase (CRL3)[34]. Structurally, N-terminal meprin and TRAF homology (MATH) domain of SPOP is responsible for substrate binding, whereas a C-terminal BR-C, ttk, and bab (BTB) domain mediates its dimerization as well as the interaction with cullin-3[35,36]. The role of SPOP in tumorigenesis is tumor type-dependent, but it primarily acts as a tumor suppressor[37]. Mechanistic studies demonstrated that SPOP targets substrate for ubiquitylation-dependent degradation or non-degradative ubiquitylation to regulate many cellular events[36,38,39]. Several SPOP substrates, including AR (androgen receptor), SRC3s (steroid receptor co-activator 3), FASN (fatty acid synthase), 17β-HSD4 (17β-hydroxysteroid dehydrogenase types 4), and ILF3 (interleukin enhancer-binding factor 3), are known to regulate cellular metabolism[40–45]. Whether and how CRL3^SPOP regulate glutamine metabolism is completely unknown.

In this study, we reported that neddylation inhibitor MLN4924 promotes glutamine metabolism by inactivating CRL3^SPOP to enhance glutamine uptake. Mechanistically, ASCT2, a major glutamine transporter, is a substrate of CRL3^SPOP, which is accumulated upon MLN4924 treatment to increase glutamine uptake. Interestingly, glutamine deprivation itself triggers SPOP self-ubiquitylation and degradation, leading to ASCT2 accumulation. Therapeutically, the combination of MLN4924 with glutamine metabolism inhibitor V-9302 markedly enhanced the killing of breast cancer cells both in vitro and in vivo. In breast cancer tissues, an inverse correlation was found between SPOP and ASCT2, and breast cancer tissues have enhanced glutamine metabolism. Taken together, our study established a mechanism by which neddylation regulates glutamine metabolism via the SPOP-ASCT2 axis, and provided a rational drug combination for enhanced anticancer efficacy.

## Results

**Neddylation blockage promotes glutamine metabolism through ASCT2.** We have recently shown that neddylation inhibition by small molecule inhibitor MLN4924, also known as pevonedistat, remarkably changed global metabolism, including impaired mitochondrial function and activated glycolysis[29,31]. Here metabolic profiling was used to identify differentially expressed metabolites, followed by pathway analysis to precisely define the altered metabolisms. In both cell extracts and culture supernatants, the glutamine metabolism was significantly affected by MLN4924 with a pathway impact value of ~0.7 (Fig. 1a, b). Specifically, MLN4924 significantly reduced the levels of glutamine and glutamate in the cell extracts, and the ratio of glutamine/glutamate in the culture supernatants (Supplementary Fig. 1a–c), the typical biomarkers for an increased glutamine metabolism[46].

Following this lead, we found that in two breast cancer cell lines MDA-MB-231 and BT549, MLN4924 substantially increased in a dose-dependent manner (a) glutamine uptake (as evidenced by reduced glutamine levels in cultural medium) and (b) medium glutamate levels (Fig. 1c, d, Supplementary Fig. 1d, e). Importantly, under neddylation blocked condition, glutamine starvation significantly inhibits cell survival in a dose dependent manner (Fig. 1e, Supplementary Fig. 1f), suggesting that neddylation regulates glutamine metabolism and consequently affecting cell survival.

To elucidate the molecular mechanism by which MLN4924 affects the glutamine metabolism, we measured the levels of several primary transporters, including ASCT2, SNAT1 and SNAT2 and few enzymes such as GLS, GOT2 and GLUD1

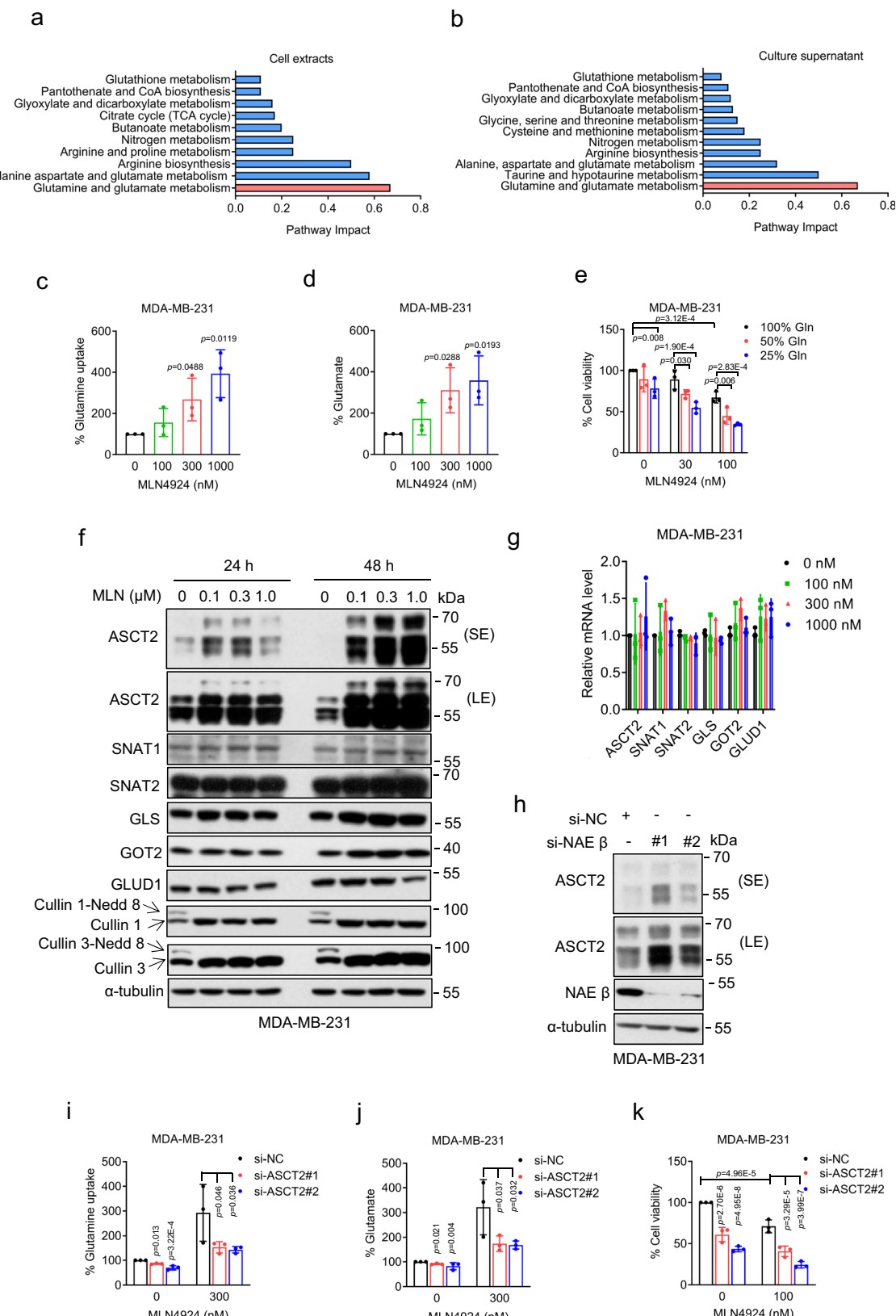

involved in glutamine metabolism. MLN4924, by effectively inhibiting neddylation of cullin 1 and cullin 3 (as positive controls), increased the levels of ASCT2 protein in time- and dose-dependent manners, but without affecting the levels of other proteins tested (Fig. 1f, Supplementary Fig. 1g), although it also increased their mRNA levels within a range of 1.5- to 2-fold

indirectly (Fig. 1g, Supplementary Fig. 1h). We noticed that multiple ASCT2 bands were detected by ASCT2 Ab (Fig. 1f) in MDA-MB231 cells, which is likely due to N-linked glycosylation as previously reported[47]. Indeed, the treatment of cell lysates of several breast and lung cancer cell lines with N-glycosidase F (PNGase F) to remove the glycosylation led to detection of single

**Fig. 1 MLN4924 promotes glutamine metabolism through ASCT2. a, b** MDA-MB-231 cells were treated without or with 300 nM MLN4924 for 24 h and subjected to untargeted metabolic profiling and integrated pathway analysis. The pathways of markedly changed metabolites in cell extracts (**a**) and culture supernatant (**b**) were shown. **c, d** MDA-MB-231 cells were treated with MLN4924 for 24 h, and the culture media were collected to analyze glutamine uptake (**c**) and glutamate production (**d**) (mean ± SD, $n = 3$). **e** MDA-MB-231 cells were treated with MLN4924 under various percentage of glutamine in culture media for 48 h, and the number of viable cells was quantified by trypan blue exclusion assay (mean ± SD, $n = 3$). **f** MDA-MB-231 cells were treated with MLN4924 for 24 h or 48 h and analyzed by immunoblotting. **g** MDA-MB-231 cells were treated with MLN4924 for 24 h, followed by qRT-PCR analysis (mean ± SD, $n = 3$). $p > 0.05$. **h** MDA-MB-231 cells were transfected with siRNA targeting NAEβ or scramble control siRNA for 48 h, followed by immunoblotting. **i, j** MDA-MB-231 cells were transfected with indicated siRNAs (si-NC or si-ASCT2) against ASCT2 for 24 h, then treated with 300 nM MLN4934, followed by glutamine uptake (**i**) and glutamate production (**j**) detection after 24 h (mean ± SD, $n = 3$). **k** MDA-MB-231 cells were transfected with scramble control siRNA (si-NC) or siRNA targeting ASCT2 (si-ASCT2) and then treated with DMSO control or 100 nM MLN4924, followed by trypan blue exclusion assay for cell viability after 48 h (mean ± SD, $n = 3$). Two-tailed, unpaired, $t$ test for **c, d**. One-way ANOVA/LSD test for **e, g, i, j, k**. Source data are provided as a Source Data file.

unmodified ASCT2 band with expected size (Supplementary Fig. 1i). We further used genetic approach to examine the effect of depletion of NAEβ, the catalytic subunit of NAE to which MLN4924 binds to and inhibits, and found that the levels of ASCT2 also increased upon NAEβ knockdown (Fig. 1h, Supplementary Fig. 1j), thus excluding the possibility that MLN4924-induced ASCT2 accumulation is an off-target effect.

We next investigated possible causal role of ASCT2 in MLN4924-induced glutamine metabolism by a rescue experiment. ASCT2 knockdown (Supplementary Fig. 1k) reduced the glutamine uptake (Fig. 1i, Supplementary Fig. 1l) and glutamate levels induced by MLN4924 (Fig. 1j, Supplementary Fig. 1m), respectively. Moreover, ASCT2 knockdown further enhanced cell killing by MLN4924 (Fig. 1k, Supplementary Fig. 1n). Taken together, these results demonstrated that MLN4924 enhanced glutamine metabolism via increasing the levels of glutamine transporter ASCT2 through inactivating neddylation E1 NAE, leading to an enhanced cell killing, induced by glutamine starvation.

**SPOP interacts with ASCT2 via its consensus degron motif.** Given that neddylation inhibition by MLN4924 inactivates CRL activity and causes the accumulation of CRL substrates[32], we hypothesized that ASCT2 could be a substrate of CRL. To this end, we first knocked down cullins 1 to 5 individually, and found that the knockdown of cullin 1 and cullin 3, but not of other cullins, caused ASCT2 accumulation significantly (Fig. 2a, Supplementary Fig. 2a, b). We first attempted to identify the F-box proteins, the receptor component of CRL1/SCF E3, that bind to ASCT2 by pull-down assay and found that among 9 F-box proteins tested, FBXL7, FBXO4 and SKP2 indeed pulled-down ASCT2 (Supplementary Fig. 2c). However, ectopic expression of these three F-box proteins individually did not change ASCT2 levels (Supplementary Fig. 2d).

We then focused our attention to cullin 3-based CRL3. Analysis of the ASCT2 protein sequence revealed an evolutionary conserved SPOP binding consensus (SBC) motif on $^{349}$GTSSS$^{353}$ (φ-π-S-S/T-S/T; φ, nonpolar residues; π, polar residues)[34] (Fig. 2b). It is well-established that the MATH domain and BTB domain of SPOP is responsible for substrate binding and cullin 3 interaction, respectively, and tumor-derived somatic mutations are largely clustered within MATH domain of SPOP, which impair the interaction of SPOP with its substrates[34] (Fig. 2c). We next determined whether SPOP indeed bind to ASCT2 and if so, by which domain. Under physiological condition, SPOP and ASCT2 bind to each other at the endogenous levels (Fig. 2d). Reciprocal immunoprecipitations showed that while exogenously expressed wild-type (WT) SPOP and ASCT2 interacts with each other, the SPOP mutants with the deletion of MATH domain, but not of BTB domain, completely abrogated ASCT2 binding (Fig. 2e, f). Consistently, an ASCT2

mutant with the SPOP binding motif/degron deleted (ASCT2ΔGTSSS) also failed to bind with SPOP (Fig. 2g).

We searched the COSMIC database (https://cancer.sanger.ac.uk/cosmic), which revealed SPOP mutations in 94 breast cancer tissues out of 5798 tested samples, with the highest frequent mutant (E78K), detected in 8 cases of breast cancer tissues, on MATH domain (Fig. 2c, Supplementary Fig. 2e). We characterized this E78K mutant, and found it had significantly reduced binding with ASCT2 (Fig. 2h). Collectively, these results showed that the SPOP-ASCT2 binding requires both MATH domain on SPOP and SPOP recognizing motif on ASCT2.

**SPOP negatively regulates ASCT2 stability by promoting its ubiquitylation.** Having established that SPOP interacts with ASCT2, we next determined the effect of SPOP manipulations on ASCT2 levels. Ectopic expression of WT SPOP, but not its mutants, including SPOPΔBTB, SPOPΔMATH, or SPOP-E78K, reduced the levels of endogenous ASCT2 in a dose-dependent manner (Fig. 3a, Supplementary Fig. 3a). Consistently, SPOP knockdown caused accumulation of ASCT2 in both breast (Fig. 3b, Supplementary Fig. 3b) and lung (Supplementary Fig. 3c) cancer cell lines, respectively, and markedly extended ASCT2 protein half-life (Fig. 3c, d, Supplementary Fig 3d–f). Thus, the stability of ASCT2 is negatively regulated by SPOP.

We then determined whether SPOP promoted the ubiquitylation of ASCT2. Indeed, wild-type SPOP, but not its various mutants promoted ASCT2 polyubiquitylation (Fig. 3e, f). Moreover, SPOP only promoted the polyubiquitylation of wild-type ASCT2, but not its degron deleted mutant (ASCT2ΔGTSSS) (Fig. 3g). Finally, using various ubiquitin mutants, we found that SPOP-mediated ASCT2 polyubiquitylation is via the K48 linkage for targeted degradation (Fig. 3h, Supplementary 3g). Together, these results showed that SPOP shortens ASCT2 protein half-life by promoting its ubiquitylation for subsequent proteasome degradation. Thus, ASCT2 is a bona fide substrate of CRL3$^{SPOP}$ E3 ligase.

**CK1δ facilitates the SPOP-ASCT2 binding and ASCT2 ubiquitylation.** It has been reported that phosphorylation of the serine residues of the substrate-binding consensus (SBC) motif is critical for SPOP binding[43,48,49]. To determine whether serine phosphorylation on the binding motif of ASCT2 affects SPOP binding, we made a ASCT2 mutant with replacement of all three serine residues to alanine (GTSSS to GTAAA, designated as ASCT2-3A) or serine to aspartic acid (GTSSS to GTDDD, designated as ASCT2-3D), and found that while the ASCT2-3A mutant almost completely lost its binding with SPOP, the ASCT2-3D mutant showed an enhanced binding with SPOP (Fig. 4a), supporting that phosphorylation on the serine residues of ASCT2 SBC motif is crucial for SPOP biding.

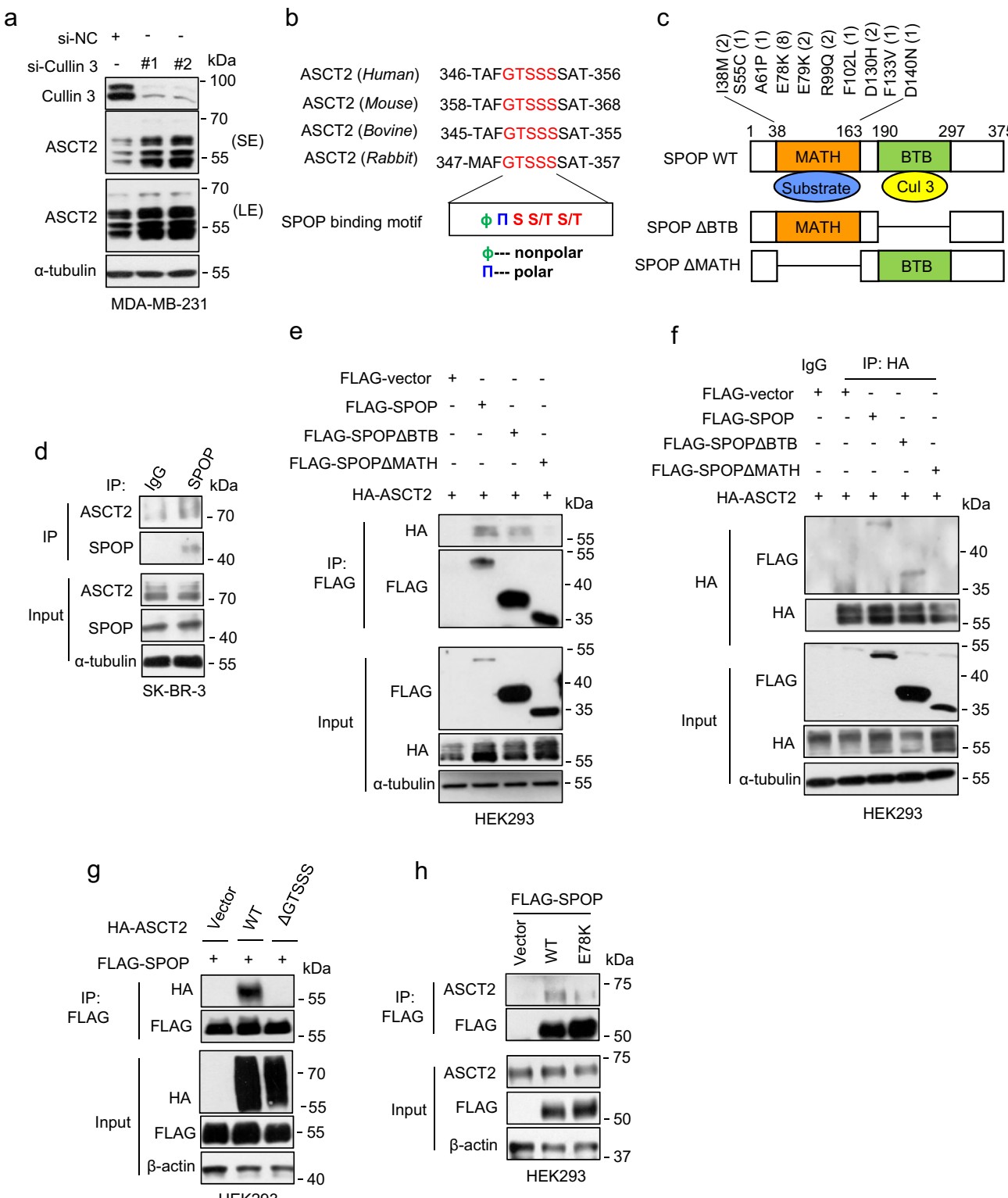

**Fig. 2 SPOP interacts with ASCT2. a** MDA-MB-231 cells were transfected siRNA targeting Cullin 3, followed by immunoblotting. **b** Evolutionary conservation of SPOP degron motif on ASCT2. **c** Schematic diagram of SPOP domains. Shown are breast cancer-associated mutations and cases in MATH domain. **d** Endogenous SPOP protein in SK-BR-3 cell lysates were pulled down with anti-SPOP antibody, and associated ASCT2 were detected by immunoblotting with anti-ASCT2 Ab (top, IP). Cell lysates were subjected to immunoblotting (bottom, input). **e, f** HEK293 cells were transfected with indicated plasmids, immunoprecipitated with FLAG-agarose beads (**e**) or HA antibody (**f**) and analyzed by immunoblotting. **g** HEK293 cells were transfected with FLAG-SPOP and indicated HA-ASCT2 plasmids, immunoprecipitated with FLAG-agarose beads and analyzed by immunoblotting. **h** HEK293 cells were transfected with plasmids expressing either wild type (WT) indicated FLAG-SPOP or E78K mutant, followed by immunoprecipitation with FLAG-agarose beads and analyzed by immunoblotting. Source data are provided as a Source Data file.

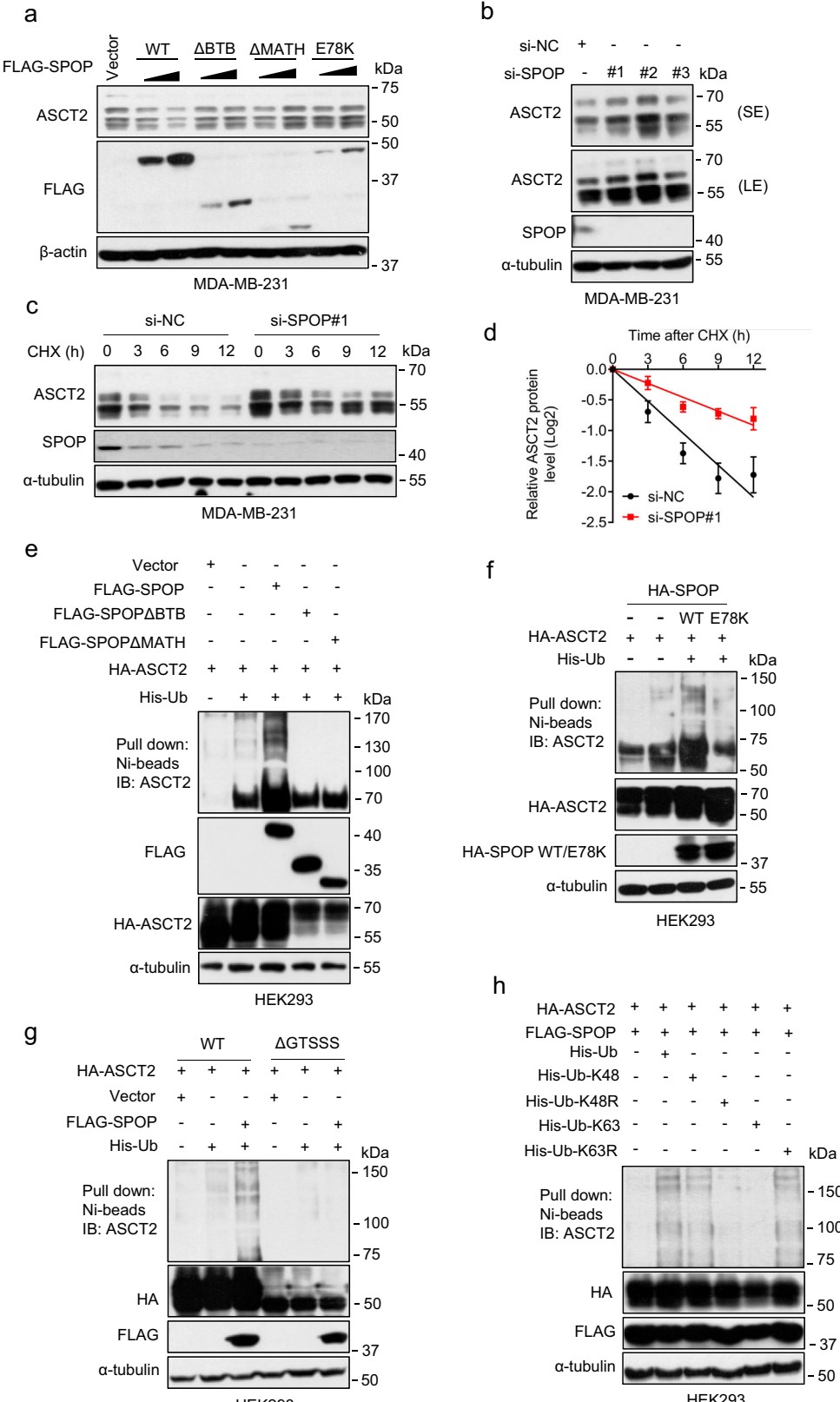

**Fig. 3 SPOP shortens ASCT2 protein half-life and promotes ASCT2 ubiquitylation. a** MDA-MB-231 cells were transfected with FLAG-SPOP and various mutants for 48 h, followed by immunoblotting. **b** MDA-MB-231 cells were transfected siRNAs targeting SPOP for 48 h, followed by immunoblotting. **c, d** MDA-MB-231 cells were transfected with siRNA targeting SPOP (si-SPOP#1) or scramble control siRNA for 48 h, then incubated with cycloheximide (CHX). Cells were harvested at indicated time points for immunoblotting (**c**). The band density of ASCT2 was quantified using ImageJ software and normalized to α-tubulin (mean ± SD, $n = 3$) (**d**). **e–h** HEK293 cells were transfected with indicated SPOP plasmids (**e**), indicated ASCT2 plasmids (**f**), or with indicated ubiquitin mutants (**h**) lysed under denaturing conditions, followed by Ni-beads pulldown, and immunoblotting for ASCT2 (top panels). The bottom panels are direct immunoblotting using whole cell extracts. Source data are provided as a Source Data file.

We then used the computer-aided algorithms (GPS 3.0 http://gps.biocuckoo.org) to search for possible kinase responsible for ASCT2 phosphorylation on the serine residues at the SBC motif, and identified CK1 as the candidate kinase. We first found that CK1δ, but not its family members CK1α or CK1ε specifically interacted with ASCT2 (Fig. 4b). Moreover, the interaction of CK1δ and ASCT2 at the endogenous levels was detected through a pull-down assay (Fig. 4c). While ectopic expression of CK1δ, but not CK1α or CK1ε reduced ASCT2 level (Fig. 4d, Supplementary Fig. 4a), CK1δ knockdown caused ASCT2 accumulation (Fig. 4e, Supplementary Fig. 4b). Furthermore, D4476, a small molecule inhibitor of CK1, abrogated ASCT2 binding with SPOP (Fig. 4f), and caused ASCT2 accumulation in a dose dependent manner (Fig. 4g, Supplementary Fig. 4c). Consistently, CK1δ knockdown extended ASCT2 protein half-life (Fig. 4h, i, Supplementary Fig. 4d-i). Finally, CK1δ enhanced SPOP-induced polyubiquitylation of wild-type ASCT2, while SPOP/CK1δ failed to promoted polyubiquitylation of the ASCT2-3A mutant (Fig. 4j). These results demonstrated that CK1δ acts as an upstream modifying kinase to phosphorylate serine residues within the SPOP binding consensus motif of ASCT2 to facilitate the SPOP-ASCT2 binding and subsequent ASCT2 ubiquitylation.

**Glutamine regulation of the SPOP-ASCT2 axis**. We next investigated possible physiological or stressed conditions that would modulate the SPOP-ASCT2 axis. A previous study showed that SPOP is subjected to upregulation by hypoxia[50]. We found that while chemical hypoxia induced by $CoCl_2$ indeed caused a time-dependent increase of SPOP, it had minimal, if any, effect on the levels of ASCT2 (Supplementary Fig. 5a), excluding possible hypoxia regulation. We also found that neither the SPOP-ASCT2 axis, nor NEDD8 and few neddylation enzymes, including E1 heterodimer NAE1/APPBP1, and UBA3/NAEβ, and two E2s, UBE2M and UBE2F is subjected to regulation by glucose deprivation in breast cancer cells (Supplementary Fig. 5b, c).

We next focused on glutamine whose cellular transportation is mainly dependent of ASCT2. Interestingly, as the glutamine concentrations in the media reduced, we observed a dose-response increase of ASCT2, but dose-dependent decrease of SPOP and CK1δ (Fig. 5a, Supplementary Fig. 5d) Moreover, glutamine deprivation caused the increase of ASCT2 and the decrease of SPOP and CK1δ in a time dependent manner (Fig. 5b, Supplementary Fig. 5e), which can be largely rescued upon glutamine addition also in a time dependent manner (Supplementary Fig. 5f, g). Thus, the levels of SPOP and ASCT2 were inversely correlated in a manner dependent of glutamine concentration in the culture medium. Out of the curiosity, we found that in breast cancer cells glutamine deprivation had no effect on the levels of NEDD8 and few neddylation enzymes, including E1 heterodimer NAE1/APPBP1, and UBA3/NAEβ, and two E2s, UBE2M and UBE2F (Supplementary Fig. 5h, i), whereas ectopic expression of NEDD8, UBE2M or UBE2F had no effect on the levels of SPOP or ASCT2 (Supplementary Fig. 5j, k).

We next investigated glutamine regulation of SPOP, using glutamine-free medium, which caused complete elimination of SPOP (Fig. 5a). Indeed, glutamine deprivation reduced the levels of both endogenous and exogenous SPOP, which are blocked by proteasome inhibitor MG132, but not by autophagy inhibitor CQ (Fig. 5c, d, Supplementary Fig. 6a, b). Furthermore, glutamine deprivation shortened the SPOP protein half-life (Fig. 5e, f, Supplementary Fig. 6c, d), and significantly enhanced WT SPOP self-ubiquitylation (Fig. 5g, Supplementary Fig. 6e). Although a ligase-dead SPOPΔBTB mutant appears to be slightly polyubiquitylated, it is independent of glutamine deprivation (Supplementary Fig. 6e). Taken together, these results clearly

demonstrated that the SPOP stability is subjected to glutamine regulation, and glutamine deprivation triggers SPOP self-ubiquitylation and subsequent proteasome degradation. We also excluded the possibility that SPOP is subjected to neddylation (Supplementary Fig. 6f), thus glutamine-mediated SPOP self-ubiquitylation is not subjected to neddylation regulation.

We further explored the underlying mechanism by which glutamine deprivation triggers SPOP self-ubiquitylation. SPOP was reported to be subjected to dimerization, which enhanced its cullin 3 binding and CRL3^SPOP E3 ligase activity[34,36]. We, therefore, determined whether glutamine withdrawal affected SPOP dimerization and found that glutamine deprivation indeed decreased SPOP dimerization (Fig. 5h), as well as the SPOP-cullin 3 binding (Supplementary Fig. 6g). Moreover, glutamine deprivation decreased the interaction of SPOP and ASCT2 at endogenous levels (Supplementary Fig. 6h) and inhibited SPOP-induced ASCT2 polyubiquitylation (Fig. 5i). Taken together, glutamine deprivation inhibited SPOP dimerization to trigger SPOP self-ubiquitylation, and reduced SPOP-ASCT2 binding, leading to reduced CRL3^SPOP E3 ligase activity toward ASCT2.

Previous studies showed that serine phosphorylation by MAPK of FBXW7, a F-box protein, regulates its dimerization and self-ubiquitylation[51,52]. We, therefore, determined whether SPOP is also subjected to such a regulation upon glutamine deprivation. We used a computer-aided algorithms (GPS 3.0 http://gps.biocuckoo.org) to search for the possible kinase that would phosphorylate the serine residues on BTB domain of SPOP, responsible for SPOP dimerization. We identified GRK2 (G protein-coupled receptor kinase) as the top candidate kinase that may phosphorylate serine residue at codon 222 (SPOP^S222). We first showed that SPOP indeed interacted with GRK2 at the endogenous levels (Fig. 5j), and glutamine deprivation enhanced their interaction (Supplementary Fig. 6h, i). We then used pharmacological and genetic approaches to inactivate or deplete GRK2 and found that paroxetine (PX), a GRK2 small molecule inhibitor[53], or siRNA-based GRK2 knockdown rescued SPOP reduction induced by glutamine-deprivation (Fig. 5k, l and Supplementary Fig. 6j, k), suggesting that GRK2 is likely a modifying kinase that in response to glutamine deprivation phosphorylates SPOP to disrupt SPOP dimerization and trigger SPOP self-ubiquitylation and degradation.

To further confirm this, we made SPOP^pS222 (SPOP-phosphor-Ser222) antibody and found that glutamine deprivation indeed induced SPOP phosphorylation at the Ser^222 in a time dependent manner (Fig. 5m, Supplementary Fig. 6l). Furthermore, GRK2 overexpression enhanced SPOP phosphorylation on Ser222 (Fig. 5n, Supplementary Fig. 6m). We also generated two SPOP mutants on codon 222 from serine to alanine (designated as S222A with phosphorylation-site abrogated) or serine to aspartic acid (designated as S222D, mimicking constitutively phosphorylation), and made the following observations: (1) compared to wild type SPOP, the S222A mutant had a prolonged protein half-life, while the S222D mutant had a shorter one (Supplementary Fig. 6n, o); (2) the S222A mutant formed stronger dimer, whereas the S222D mutant formed the weaker one, which was more obviously seen under glutamine deprivation (Fig. 5o); and (3) consistently, the S222A mutant had reduced self-ubiquitylation, whereas the S222D mutant had an increased one (Fig. 5p). Finally, we found that GRK2 inhibitor PX inhibited ASCT2 ubiquitylation by SPOP under glutamine-deprived but not glutamine-enriched condition (Fig. 5q). Taken together, these results demonstrated that in response to glutamine deprivation, GRK kinase is induced/activated to phosphorylate SPOP on the Ser^222 residue that blocks SPOP dimerization to trigger SPOP self-ubiquitylation and degradation, leading to inactivation of CRL3^SPOP E3 ligase and subsequent accumulation of ASCT2 substrate.

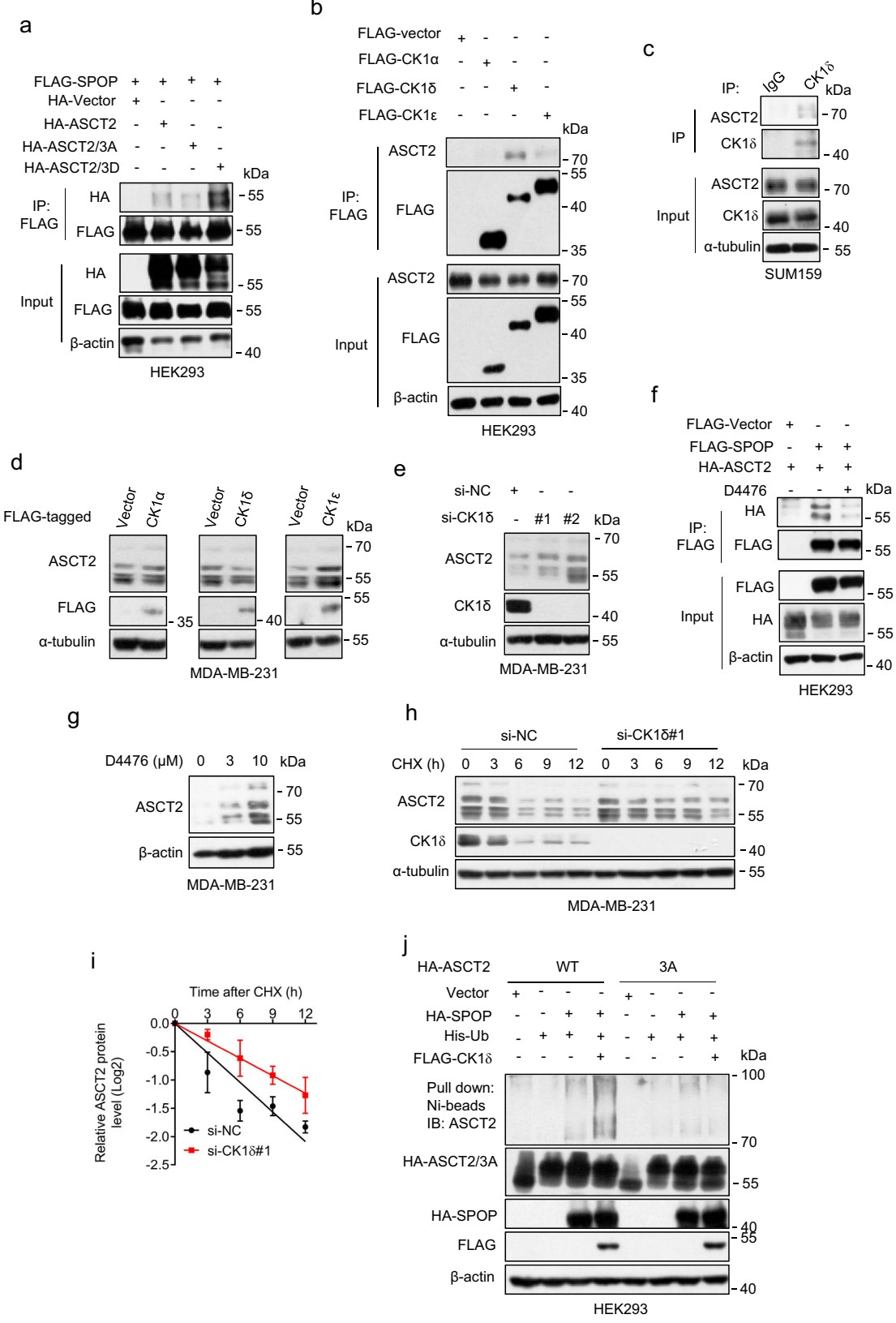

**The SPOP-ASCT2 axis regulates glutamine metabolism**. We next evaluated biologically whether the SPOP-ASCT2 axis regulates glutamine metabolism. While ASCT2 knockdown (Fig. 6a) reduced cellular glutamine uptake and glutamate production as expected, SPOP knockdown (Fig. 6a) increased both. Significantly, this enhancing effect of SPOP knockdown was completely rescued

by simultaneous ASCT2 knockdown (Fig. 6b, c, Supplementary Fig. 7a, b). Consistently, compared with wild-type ASCT2, ectopic expression of ASCT2-3A, a SPOP-resistant mutant (Supplementary Fig. 7c), enhanced cellular glutamine uptake and glutamate production (Fig. 6d, e, Supplementary Fig. 7d, e). Thus, the promotion of glutamine metabolism by MLN4924-mediated

**Fig. 4 CK1δ is required for SPOP-mediated ASCT2 degradation. a** HEK293 cells were transfected with indicated plasmids, followed by immunoprecipitation with FLAG-agarose beads and immunoblotting. **b** HEK293 cells were transfected with various FLAG-CK1 isoforms, immunoprecipitated with FLAG-agarose beads and analyzed by immunoblotting. **c** Endogenous CK1δ protein in SUM159 cell lysates were pulled down with anti-S CK1δ antibody, and associated ASCT2 were detected by immunoblotting with anti-ASCT2 Ab (top, IP). Cell lysates were subjected to immunoblotting (bottom, input). **d** MDA-MB-231 cells were transfected with indicated CK1 isoforms, then analyzed by immunoblotting. **e** MDA-MB-231 cells were transfected siRNA targeting CK1δ for 48 h, followed by immunoblotting. **f** HEK293 cells were transfected with FLAG-SPOP and HA-ASCT2 and treated with D4476 for 12 h before harvesting, followed by immunoprecipitation with FLAG-agarose beads and analyzed by immunoblotting. **g** MDA-MB-231 cells were treated with different concentrations of CK1 inhibitor D4476 for 24 h, then analyzed by immunoblotting. **h, i** MDA-MB-231 cells were transfected with si-NC or siRNA targeting CK1δ (si-CK1δ#1), then incubated with CHX for various time points, and analyzed by immunoblotting ies (**h**), the band density of ASCT2 was quantified using ImageJ software and normalized to α-tubulin (mean ± SD, n = 3) (**i**). **j** HEK293 cells were transfected with indicated plasmids, lysed under denaturing conditions, followed by Ni-beads pulldown, and immunoblotting for ASCT2. Source data are provided as a Source Data file.

neddylation blockage is through, at least in part, the inactivation of CRL3$^{SPOP}$ E3 ligase and subsequent accumulation of ASCT2. We also directly compared ASCT2 levels upon ectopic expression vs. MLN4924 treatment, and found that ASCT2 levels are higher under overexpression condition than that induced by MLN4924 treatment (Supplementary Fig. 7f). The observation that MLN4924 is more efficient in promoting glutamine uptake and glutamate production than that of HA-ASCT2 overexpression (Fig. 1c, d, Supplementary Fig. 1d, e vs. Figure 6d, e, Supplemental Fig. 7d, e), suggesting that MLN4924 may also has ASCT2-independent effect in modulating glutamine metabolism.

**The SPOP-ASCT2 axis regulates growth and survival of breast cancer cells.** We next investigated the effect of the SPOP-ASCT2 axis on growth and survival of breast cancer cells. Knockdown of ASCT2 or SPOP markedly inhibited or promoted cell growth and clonogenic survival, respectively, whereas co-knockdown of ASCT2 and SPOP also caused growth suppression (Fig. 6f, g, Supplementary Fig. 8a, b), suggesting that while SPOP acts as a growth suppressor, ASCT2 plays the major role for the growth and survival of breast cancer cells.

We then extended this observation to the in vivo xenograft tumor model derived from MDA-MB-231 cells after shRNA-based knockdown of ASCT2 or SPOP or both (Fig. 6h) and found that consistent with cell culture model, ASCT2 knockdown inhibited, whereas SPOP knockdown promoted tumor growth. Co-knockdown of both also inhibited tumor growth (Fig. 6i–k), demonstrating a dominant role of ASCT2 in controlling of in vivo tumor growth. We also confirmed that ASCT2 or SPOP were indeed knocked down in respective tumor tissues (Fig. 6l). Taken together, ASCT2 appears to be a strong oncogenic protein, whereas SPOP is relatively a weak tumor suppressor in breast cancer cells.

**Enhanced cancer killing by co-targeting neddylation and ASCT2.** What is the translational implication of our finding? Given that neddylation inhibition promotes glutamine metabolism by causing ASCT2 accumulation via inactivation of CRL3$^{SPOP}$, we next determined whether the newly identified ASCT2 inhibitor, V-9302[18], would sensitize breast cancer cells to MLN4924. We measured the IC50 values of V-9302 in two breast cancer cell lines (Supplementary Fig. 8c, d), and used the low toxic IC$_{20}$ values to determine MLN4924 sensitization. Indeed, V-9302 increased the sensitivity of breast cancer cells to MLN4924 in both tested concentrations (Fig. 6m, Supplementary Fig. 8e). Moreover, the combination of V-9302 and MLN4924 caused greater inhibition of clonogenic survival than either of compound used alone (Fig. 6n, Supplementary Fig. 8f). Similarly, l-γ-glutamyl-p-nitroanilide (GPNA), a potent ASCT2 inhibitor[17], also sensitized breast cancer cells to MLN4924, as measured by both proliferation and clonogenic survival assays (Supplementary Fig. 8g–j).

Finally, we used an in vivo xenograft tumor model of MDA-MB-231 cells to evaluate the anticancer efficiency of MLN4924, or V-9302 alone or in combination at nontoxic doses (Supplementary Fig. 8k). Compared with vehicle control, subcutaneous tumor growth was inhibited by MLN4924 or V-9302 alone, and further inhibited by combination of MLN4924 with V-9302 (Fig. 6o). Consistently, the tumor weight at the end of the experiment also showed a remarkable reduction in the treatment groups, particularly with combination of MLN4924 and V-9302, as compared to the control group (Fig. 6p). Together, our results showed that MLN4924-induced accumulation of ASCT2 appears to be a "side-effect" of the compound, which can be overcome by ASCT2 inhibitor, providing a rational drug combination for maximal therapeutic efficacy in cancer treatment.

**Inversed correlation between SPOP and ASCT2, and enhanced glutamine metabolism in breast cancer tissues.** To determine expression correlation between SPOP and ASCT2, we used immunohistochemistry staining to measure the levels of SPOP and ASCT proteins in human breast cancer tissues, and their correlations with patient survival. The inversed protein levels were found between ASCT2 and SPOP in both non-tumor and tumor tissues (Fig. 7a, b, Supplementary Fig. 9a). Compared to the adjacent normal tissues (n = 21), tumor tissues (n = 314) showed significant lower SPOP, but higher ASCT2 levels (Fig. 7c, d). Moreover, in 109 of SPOP low samples, less than 20% had low ASCT2, whereas in 205 of SPOP high samples, about 40% had low ASCT2. The difference is statistically significant (Fig. 7e), indicating an inverse relationship.

We further performed the Kaplan-Meier survival analysis among 92 cancer samples with the inverse correlation between the levels of SPOP and ASCT2, and found high SPOP/low ASCT2 (n = 39) predicted a better patient survival, whereas low SPOP/high ASCT2 (n = 53) was associated with worse survival of breast cancer patients (Fig. 7f). Thus, it appears that the levels of the SPOP-ASCT2 axis might be used to predict the survival of breast cancer patients.

Finally, we measured the glutamine metabolite levels in 12 paired breast normal vs. tumor tissues through LC-MS/MS-based targeted metabolomics analysis, and observed distinct metabolic profiles between breast normal and tumor tissues (Supplementary Fig. 9b). Compared to paired normal tissues, 10 out 12 tumor tissues had lower levels of glutamine, which is statistically significant (Fig. 7g and Supplementary Fig. 9c), whereas 9 out of 12 tumor samples had higher levels of glutamate, although it did not reach the statistically significant levels (Supplementary Fig. 9d, e). Likewise, the ratio of glutamate/glutamine was significantly higher in tumor tissues than in normal tissues (Fig. 7h, Supplementary Fig. 9f). Taken together, these results, using paired samples from the same patients, showed that cancer tissues have low or high levels of glutamine or glutamate, respectively, suggesting an enhanced glutamine metabolism. Thus,

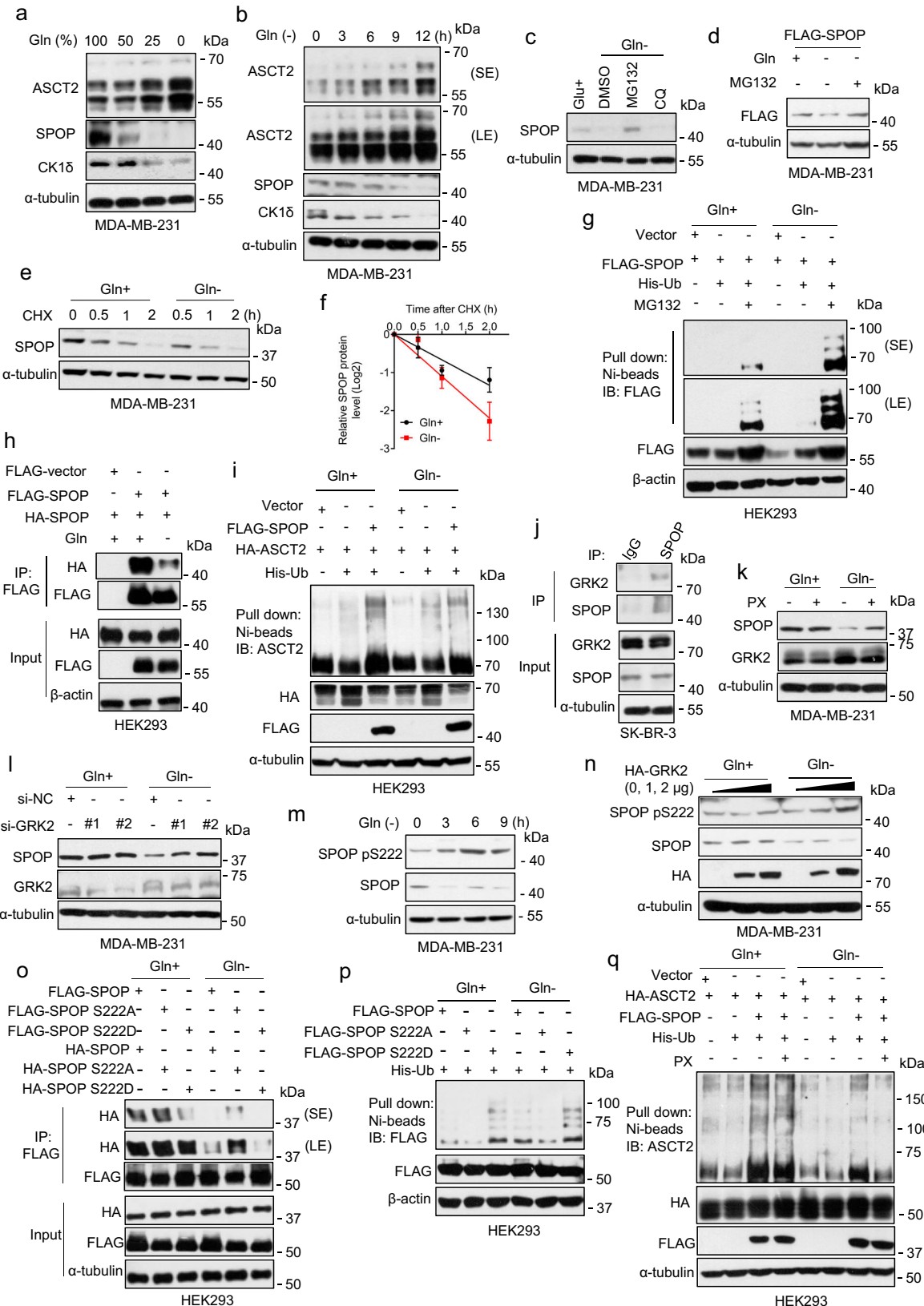

the high levels of ASCT2, coupled with the low levels of SPOP would cause an increased glutamine transportation to facilitate a high glutamine consumption in breast cancer tissues, providing mechanistic support of the notion that tumor tissues have a high glutamine consumption[8,54].

## Discussion

Our recent study showed that neddylation modification regulated mitochondrial energy metabolism and cellular glycolysis[31]. Whether or how neddylation also regulate glutamine metabolism is previously unknown. In this study, we showed that neddylation

**Fig. 5 Glutamine deprivation inhibits SPOP dimerization and promotes SPOP self-ubiquitylation. a, b** MDA-MB-231 cellswere cultured with indicated percentage of glutamine for 24 h (**a**) or glutamine-free medium at different time points (**b**), for immunoblotting. **c** MDA-MB-231 cells were cultured in glutamine-free medium for 6 h with MG132 or Chloroquine (CQ) for another 6 h or 24 h, followed by immunoblotting. **d** MDA-MB-231 cells were transfected with FLAG-SPOP plasmid, cultured in glutamine-free medium with MG132, then for immunoblotting. **e, f** MDA-MB-231 cells were cultured in glutamine-containing or glutamine-free medium with addition of CHX, harvested at indicated time points for immunoblotting. (**e**). The band density of SPOP was quantified and normalized to α-tubulin (mean ± SD, $n = 3$) (**f**). **g** HEK293 cells were transfected with indicated plasmids, cultured in glutamine-containing or glutamine-free media for 12 h with or without MG132. Cells were lysed under denaturing conditions, followed by Ni-beads pulldown, and immunoblotting with anti-FLAG antibody. **h** HEK293 cells were transfected with indicated plasmids, cultured in glutamine-containing or glutamine-free media, followed by immunoprecipitattion with FLAG-agarose beads and analyzed by immunoblotting. **i** HEK293 cells were transfected with indicated plasmids, cultured in glutamine-containing or glutamine-free media, and lysed under denaturing conditions, followed by Ni-beads pulldown, immunoblotting for ASCT2. **j** Endogenous SPOP protein in SK-BR-3 cell lysates were pulled down with anti-SPOP antibody, and associated GRK2 were detected by immunoblotting with anti-GRK2 Ab (top, IP). Cell lysates were subjected to immunoblotting (bottom, input). **k** MDA-MB-231 cells were cultured in glutamine-containing or glutamine-free media in the presence or absence of GRK2 inhibitor PX (10 μM) for 6 h, and analyzed by immunoblotting. **l** MDA-MB-231 cells were transfected with siRNAs targeting GRK2 or scramble control siRNA, then cultured in glutamine-containing or glutamine-free media for 6 h, and analyzed by immunoblotting. **m** MDA-MB-231 cells were cultured in glutamine-free medium for indicated time points and analyzed by immunoblotting. **n** MDA-MB-231 cells were transfected with indicated plasmids, then cultured in glutamine-containing or glutamine-free media for indicated time points, followed by immunoblotting. **o–q** HEK293 cells were first transfected with siRNA targeting SPOP (3'-UTR) (**o, p**) then with indicated plasmids (**o–q**). Cells were cultured with glutamine-free medium for 12 h (**o, p**) or, cultured in glutamine-containing or glutamine-free media in the absence or presence of PX for 12 h (**q**), followed by immunoprecipitation with FLAG-agarose beads and analyzed by immunoblotting (**o**) or lysed under denaturing conditions, followed by Ni-beads pulldown, and analyzed by immunoblotting for FLAG antibody (**p**) or for ASCT2 (**q**). Source data are provided as a Source Data file.

inhibition by a small molecule inhibitor, MLN4924, promotes glutamine uptake and metabolism via inhibiting ubiquitylation and degradation of glutamine transporter ASCT2 by CRL3$^{SPOP}$ E3 ligase.

Glutamine, in addition to glucose, is another main source of catabolism to fuel cancer cells for their accelerated growth and proliferation[55]. It is therefore not surprising that ASCT2, a major glutamine transporter across the cell member, is subjected to regulation at the multiple levels. Examples include upregulation by ATF4 or MYC or downregulation by RB[55–57] at the transcriptional levels, or upregulation by miR-137 at the post-transcriptional levels[58]. At the post-translational levels, ASCT2 is subjected to N-linked glycosylation modification that modulates its membrane trafficking and function[47], and is ubiquitylated and degraded by RNF5 ubiquitin ligase in response to chemotherapy-induced endoplasmic reticulum (ER) stress[59].

Herein, we showed that CRL3$^{SPOP}$ E3 ubiquitin ligase is responsible for ASCT2 ubiquitylation and degradation in response to glutamine starvation. Our conclusion is supported by the following lines of evidence: (1) inactivation of the entire neddylation or of CRL3 in particular via pharmacological or genetic approaches causes ASCT2 accumulation; (2) SPOP binds to ASCT2 under physiological conditions, and the SPOP-ASCT2 binding is dependent on the evolutionarily conserved SPOP binding motif (GTSSS); (3) SPOP ectopic expression or knockdown decreases or increases ASCT2 levels, respectively; (4) SPOP knockdown extends ASCT2 protein half-life; and (5) SPOP promotes ASCT2 ubiquitylation via the K48 linkage for degradation. We also identified CK1δ as the kinase that phosphorylates ASCT2 at the SPOP binding motif to facilitate its SPOP binding and subsequent ubiquitylation and degradation. Thus, our current study adds ASCT2 to the repertoire of the growing list of SPOP substrates[36] and extends the layers of multiple upstream regulatory mechanisms of ASCT2. We also observed up to 2-fold increase of the mRNA levels of some glutamine transporters and enzymes in BT549 cells after MLN4924 treatment (Supplementary Fig. 1h), which is likely mediated via an indirect effect, given the primary target of MLN4924 is neddylation E1 to inactivate CRLs. Furthermore, since cullin 1 knockdown also caused ASCT2 accumulation (Supplementary Fig. 2a, b), it is likely that ASCT2 is also a substrate of CRL1. How does CRL1 promote ASCT2 ubiquitylation by which F-box protein under what

physiological or stressed conditions will be an interesting project for future investigation.

Another very interesting observation we made in this study is that SPOP itself is subjected to glutamine regulation. The rapid growth of cancer cells increased glutamine utilization to cause the shortage of local glutamine supply, leading to glutamine deficiency[8,12]. To adapt this glutamine-limited microenvironment, cancer cells have developed a variety of mechanisms to maintain the proliferation and survival. The examples include inhibition of histone demethylation to promote dedifferentiation[60], transcriptional induction of Slug to promote EMT and metastasis[61] and induction of PD-L1 via activation of EGFR/ERK/c-Jun signaling to modulate immune response[62,63], promotion of acetylation-dependent degradation of glutamine synthetase via CRL4$^{CRBN}$[64], activation of p53[65–67], and induction of genomic instability[68].

Here we showed a mode of regulation in response to glutamine deprivation by directly targeting on SPOP, an E3 ligase for ASCT2, leading to enhanced ASCT2 stability. Specifically, glutamine deprivation induces/activates GRK2 kinase to phosphorylate SPOP at the Ser$^{222}$, which inhibits SPOP dimerization, leading to (a) reduced E3 ligase activity towards ASCT2 poly-ubiquitylation, and (b) enhanced SPOP self-ubiquitylation and degradation. Thus, SPOP is subjected to regulation by phosphorylation under glutamine deprivation that disrupts its self-dimerization and trigger it self-ubiquitylation and degradation. Thus, this self-sustained SPOP-ASCT2 autoregulation ensured enough glutamine being transported into cells under glutamine deficient microenvironment. A previous report showed, using mouse skin tissues, glutamine attenuates itch via a rapid induction of GRK2[69], which is in contrast to our observation that glutamine deprivation leads to GRK2 induction and activation in breast cancer cell lines. Thus, it appears that glutamine regulation of GRK2 could be in a context (tissue/cell) dependent manner. It is noteworthy that in breast cancer cells, (1) neither ASCT2 nor SPOP is subjected to glucose regulation, since glucose deprivation has no effect on their protein levels (Supplementary Fig. 5b, c); (2) glutamine deprivation has no effect on the levels of neddylation activating enzyme (NAE) E1 (including both subunits, NAE1/AAPBP1 and UBA3/NAEβ), and two E2 neddylation conjugating enzymes (UBE2M and UBE2F) (Supplementary

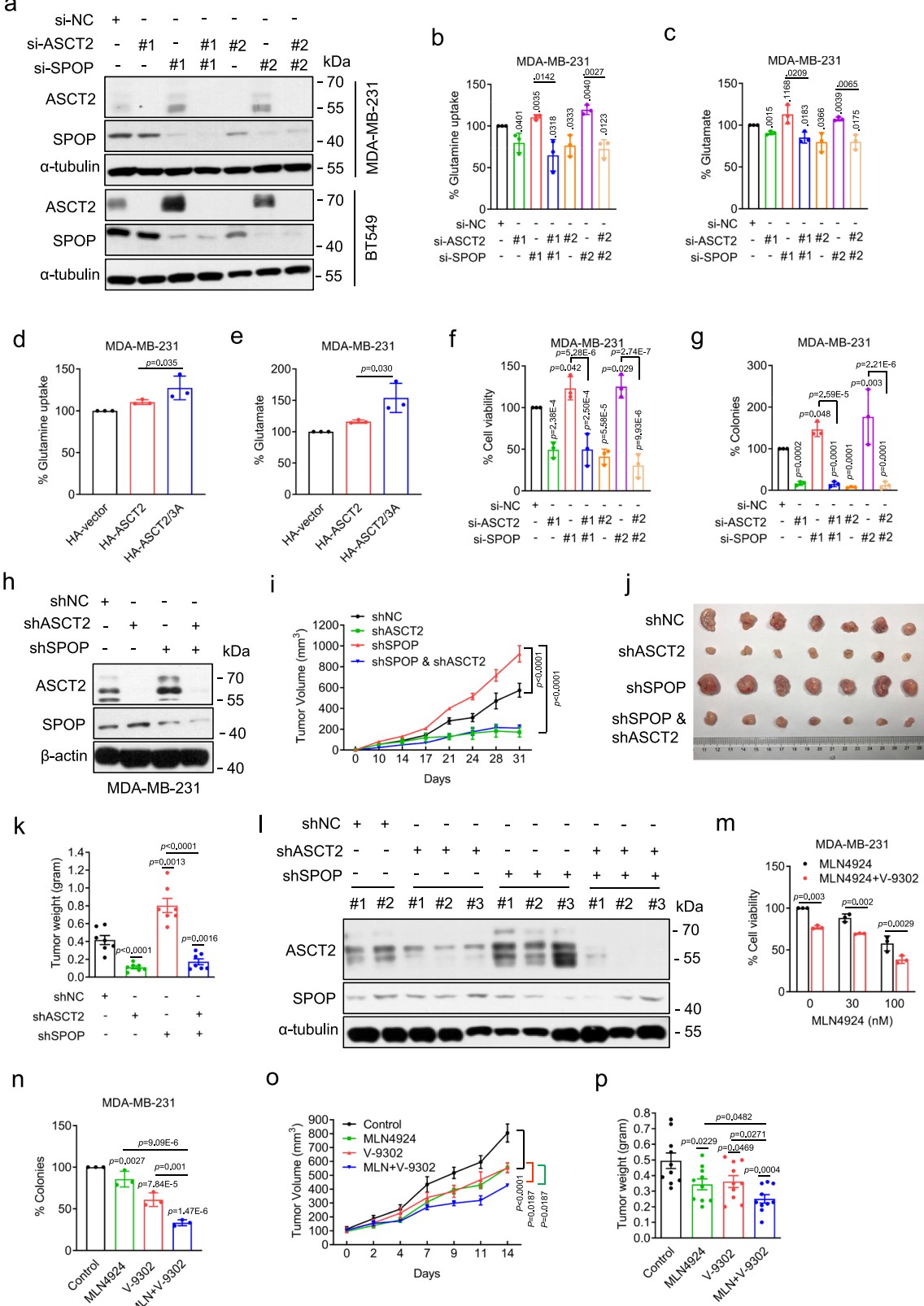

Fig. 5h, i). These results demonstrated the specificity of glutamine regulation of the SPOP-ASCT2 axis.

The finding of the SPOP-ASCT2 auto-regulation may also have translational implication. SPOP is a well-characterized tumor suppressor, whereas ASCT2 acts as an oncoprotein in most of cancer types[17,36]. In breast cancer cells, growth stimulation upon SPOP knockdown can be fully rescued by simultaneous ASCT2 knockdown seen both in vitro cell culture and in vivo xenograft tumor models, indicating a causal relationship between SPOP and ASCT2 in regulation of tumor growth. It is likely that during breast tumorigenesis, SPOP is mutated or down-regulated (Fig. 7c, Supplementary Fig. 2e) to cause ASCT2 accumulation,

**Fig. 6 The SPOP-ASCT2 axis regulates glutamine metabolism, growth and survival of breast cancer cells. a** MDA-MB-231 and BT549 cells were transfected with indicated siRNAs and subjected to immunoblotting. **b, c** MDA-MB-231 cells were transfected with indicated siRNAs for 48 h, followed by assays for glutamine uptake (**b**) and glutamate production (**c**) detection (mean ± SD, $n = 3$). **d, e** MDA-MB-231 cells were transfected with plasmids expressing wild-type ASCT2 and ASCT2/3 A mutant for 48 h, followed by assays for glutamine uptake (**d**) and glutamate production (**e**) detection (mean ± SD, $n = 3$). **f, g** MDA-MB-231 cells were transfected with indicated siRNAs and cell viability (**f**) and clonogenic survival (**g**) were measured (mean ± SD, $n = 3$). **h** MDA-MB-231 cells with stable knockdown of ASCT2, SPOP, along with the shNC control, followed by immunoblotting using indicated antibodies. **i-l** MDA-MB-231 cells with stable knockdown of ASCT2, SPOP, along with the shNC control were subcutaneously injected to nude mice individually or in combination as indicated. The in vivo tumor growth was measured and plotted (mean ± SEM, $n = 7$) (**i**). Tumors were harvested at the end of experiment, photographed (**j**), weighted (**k**), and tumor tissues were lysed, followed by immunoblotting (**l**). **m** MDA-MB-231 cells were treated with different concentrations of MLN4924 without or with 5 μM V-9302 for 72 h and cell viability was measured by trypan blue exclusion assay (mean ± SD, $n = 3$). **n** MDA-MB-231 cells were treated with 5 nM MLN4924, 5 μM V-9302, alone or in combination for 10 days for clonogenic survival (mean ± SD, $n = 3$). **o, p** The in vivo growth of MDA-MB-231 xenograft tumors after 14 days of treatment of MLN4924, or V-9302, alone or in combination, along with vehicle control. Tumor growth was monitored (mean ± SEM, $n = 10$) (**o**) and the tumor mass were weighted at the end of experiment (**p**). O-way ANOVA/LSD test for **d**, **e**, **n**, **f**, **g**. Two-tailed, unpaired, $t$ test for **b**, **c**, **k**, **m**, **p**. Two-way ANOVA/Tukey's multiple comparison tests for **i** and **o**. Source data are provided as a Source Data file.

which would increase glutamine transportation to accelerate glutamine metabolism. Consistent with the in vitro results showing that SPOP levels decreased while ASCT2 levels increased upon glutamine starvation, we found low levels of SPOP and high levels of ASCT2 were coupled with decreased glutamine concentrations and increased glutamate/glutamine ratio in tumor tissues, as compared to paired normal counterparts (Fig. 7, Supplementary Fig. 9). Thus, our study demonstrated an interesting mutual regulation of glutamine and the SPOP-ASCT2 axis, which may explain, at least in part, why some types of cancers (i.e., breast cancer) have a high glutamine consumption while glutamine deficient microenvironment modulates the adaptation mechanisms for tumor survival[60,62,63].

Furthermore, the protein levels of SPOP and ASCT2 is inversely correlated in breast cancer tissues; and lower SPOP coupled with higher ASCT2 predicts a poor patient survival, whereas higher SPOP coupled with lower ASCT2 predicts a better patient survival. Thus, the SPOP-ASCT2 axis may serve as the biomarker for prognosis of breast cancer patients, if this observation can be confirmed in much large set of samples.

What is the therapeutic significance of our study? MLN4924 is currently under several clinical trials as an anti-cancer agent, acting alone or in combination with other chemo-drugs[22]. Our recent study showed that MLN4924 promoted mitochondrial OXPHOS and increased glycolysis by activating PKM2, and combination of MLN4924 with inhibitors targeting OXPHOS or PKM2 enhanced the anti-cancer efficiency[29,31]. Here, we showed that by inactivation of CRL3[SPOP], MLN4924 caused ASCT2 accumulation to confer survival advantage by enhancing glutamine metabolism, which would counteract its therapeutic benefits. Thus, it is logical to combine MLN4924 with ASCT2 inhibitor for enhanced anti-cancer efficacy. To date, V-9302 has been identified as the best available ASCT2 inhibitor and has been used both in vitro and vivo studies[18,70,71], although an off-target effect in an oocyte expression model were reported[72]. Indeed, the combination of MLN4924 with ASCT2 inhibitor V-9302 achieved much better growth suppression of cancer cells in both in vitro cell-culture setting and in vivo xenograft tumors. Thus, this rational combination may have application against breast cancer with overexpressed ASCT2. It is noteworthy that while ASCT2 acts as a significant role for glutamine transport, it is also responsible for the transportation of alanine, serine, cysteine, and threonine[16,70,73]. Thus, MLN4924-induced ASCT2 accumulation may also alter the transportation of other amino acids to regulate their metabolic pathways, an interesting subject for further investigation.

In summary, our study fits the following working model: Under glutamine enriched condition, SPOP forms the dimer to complex with cullin 3 to promote ubiquitylation of ASCT2 for proteasome degradation. In tumors with SPOP mutation or down regulation, ASCT2 levels are increased due to reduced CRL3[SPOP] activity, leading to increased glutamine metabolism. Under glutamine deprivation condition, SPOP is phosphorylated by GRK to abrogate SPOP dimerization and trigger self-ubiquitylation, leading to ASCT2 accumulation to accelerate glutamine metabolism. The combination of MLN4924 with ASCT2 inhibitor would block enhanced glutamine metabolism to increase cancer cell killing (Fig. 7i). Our study, therefore, links the neddylation modification and glutamine metabolism, and provides a rational drug combination for enhanced anti-cancer efficacy.

## Methods

**Cell culture and transfection.** Human breast cancer cell lines MDA-MB-231 (HTB-26), BT549 (HTB-122), SK-BR-3 (HTB-30), lung cancer cell lines A549 (CCL-185), H1703 (CRL-5889), H1792 (CRL-5895) and H358 (CRL-5807), and embryonic kidney cell line HEK293 (CRL-1573) were obtained from American Type Culture Collection. Human breast cancer cell line SUM159 was a kind gift from Dr. Chenfang Dong, Zhejiang University. MDA-MB-231, BT549, H1703, H1792, and H358 cells were cultured in RPMI 1640 medium (Invitrogen), whereas SK-BR-3, SUM159, A549, and HEK293 cells were cultured in DMEM medium (Invitrogen), supplemented with 10% fetal bovine serum (Gibco) and incubated at 5% $CO_2$ incubator (Thermo) at 37°C and 95% humidity.

Cells were transfected with various plasmids using Lipofectamine 3000 Transfection Reagent or with si-RNA oligonucleotides by Lipofectamine RNAiMAX Transfection Reagent, according to the manufacturer's instructions, respectively.

**Antibodies and reagents.** The following antibodies were used: ASCT2 (Cell Signaling Technology, D7C12, 8057, dilution: 1:1000); ASCT2 (Abcam, ab84903, dilution: 1:800); β-actin (Sigma-Aldrich, A5441, dilution: 1:10000); CK1δ (Santa Cruz, sc-55553, dilution: 1:1000); CK1δ (Cell Signaling Technology, 12417 S, dilution: 1:1000); CUL-1 (Santa Cruz, sc-11384, dilution: 1:1000); CUL-2 (Abcam, ab166917, dilution: 1:1000); CUL-3 (Cell Signaling Technology, 2759 S, dilution: 1:1000); CUL-4A (Cell Signaling Technology, 2699 S, dilution: 1:1000); CUL-4B (Proteintech, 12916-1-AP, dilution: 1:1000); CUL-5 (Santa Cruz, sc-13014, dilution: 1:1000); FLAG, clone M2 (Sigma-Aldrich, F1804-500UG, dilution: 1:2000); FLAG M2 affinity gel (Sigma-Aldrich, A2220-5ML); GLUD1 (Abcam, ab89967, dilution: 1:1000); GLS (Abcepta, ap8809b, dilution: 1:1000); GOT2 (ProteinTech, 14800-1-AP, dilution: 1:1000); GRK2 (ProteinTech, 13990-1-AP, dilution: 1:1000); GRK2 (Cell Signaling Technology, 3982, dilution: 1:1000); HA (Sigma, H6908, dilution: 1:2000); Anti-HA High Affinity (3F10) (Roche, 11867423001); HIF-1α (Santa Cruz, sc-53546, dilution: 1:1000); NAEβ (Abcam, ab124728, dilution: 1:1000); NAE1 (Cell Signaling Technology, 14321S, dilution: 1:1000); NEDD8 (Abcam, ab81264, dilution: 1:1000); SNAT1 (Abcam, ab134268, dilution: 1:1000); SNAT2 (Abcam, ab90677, dilution: 1:1000); SPOP (Abcam, ab137537, dilution: 1:1000); SPOP (Affinit, DF12106, dilution: 1:800); SPOP (Santa Cruz, sc-377206, dilution: 1:1000); p-SPOP (Ser222) (dilution: 1:1000); α-tubulin (Sigma, Clone AA13, T8203, dilution: 1:10000); UBE2M (Santa Cruz, sc-390064, dilution: 1:1000); UBE2F (ProteinTech, 17056-1-AP, dilution: 1:1000); Peroxidase AffiniPure Goat Anti-Rabbit IgG (H + L) (Jackson, 11-035-144, dilution: 1:4000); Peroxidase AffiniPure Goat Anti-Mouse IgG (H + L) (Jackson, 115-035-146, dilution: 1:4000); Peroxidase AffiniPure Goat Anti-Rat IgG (H + L) (Jackson, 112-035-143, dilution: 1:4000). A peptide polyclonal Ab against phosphor-SPOP-Ser222 (AILAAR(pS)

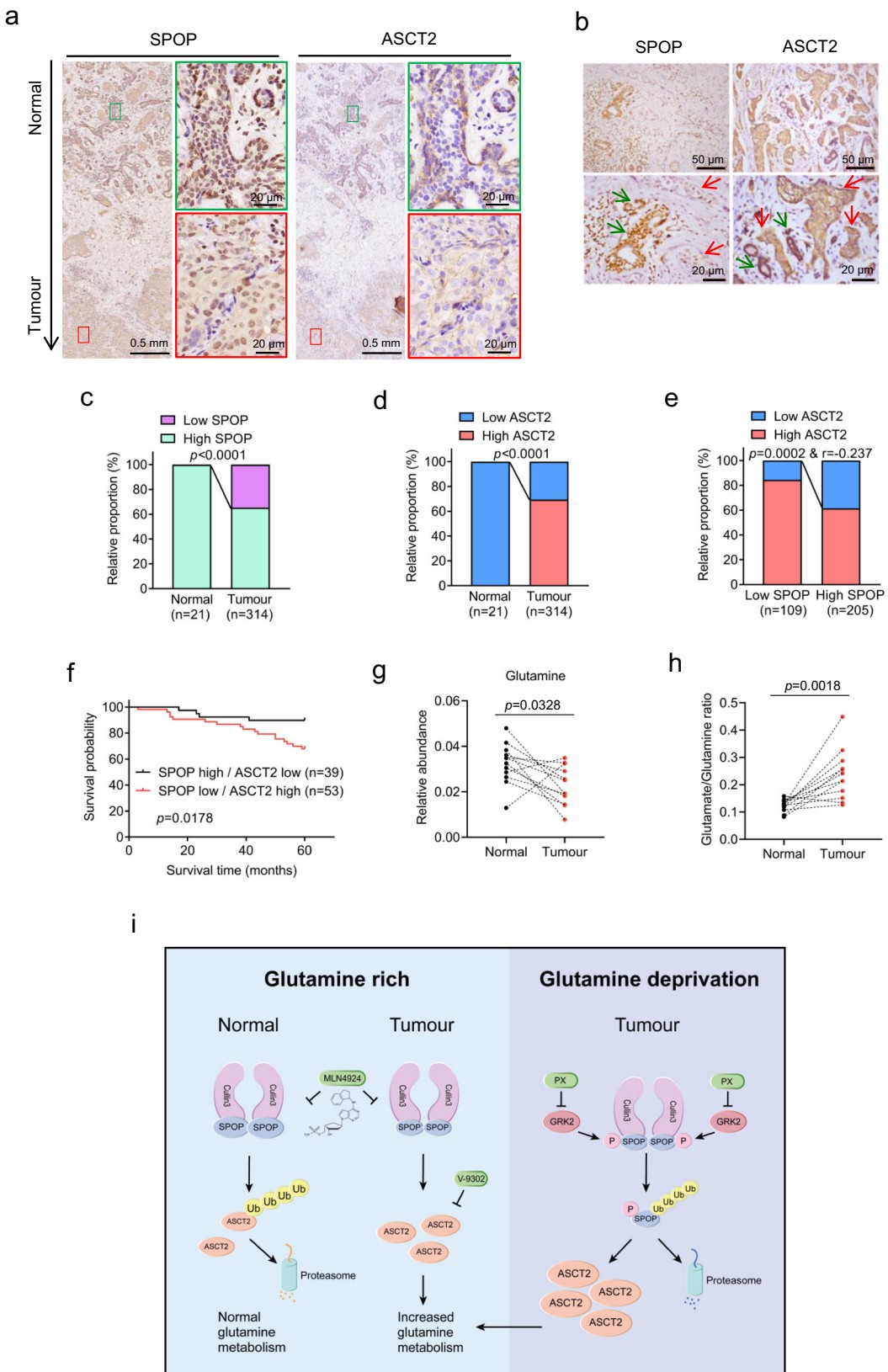

PVFSA) was generated, followed by two sequential rounds of affinity purification by Youke Biological Technology (Shanghai, China).

Reagents were obtained from the following suppliers: Chlorhexidine (CHX) (Sigma, C7698); Chloroquine (Sigma, C6628); CoCl$_2$ (Sigma, 60818-50 G); D4476 (Selleck, S7642); GPNA (Selleck, S6670); G-418 (ApexBio, A2513); MG132 (MedChem Express, HY-13259); MLN4924 (ApexBio, B1036); MLN4924 (MCE, HY-70062); Puromycin (VWR Life Science, 0336C120); Paroxetine (Selleck, S3005); PNGase F (NEB, P0704S); Lipofectamine RNAiMAX Transfection Reagent (Invitrogen, 13778-150); Lipofectamine 3000 Transfection Reagent (Invitrogen, L3000-015); Ni-NTA Agarose (Qiagen, 30210); V-9302 (Selleck, S8818).

**Plasmids, si-RNAs, and shRNAs**. The SPOP plasmids were gifts from Dr. Ping Wang (Tongji University, China), and FLAG-CK1 plasmids were gifts from Dr.

**Fig. 7 Inverse correlation of SPOP and ASCT2 levels in breast cancer tissues and their association with patient survival. a, b** Representative images of SPOP and ASCT2 staining in consecutive breast tissues (normal VS. tumor). Green box (**a**) and arrow (**b**) indicate normal region, and red box (**a**) and arrow (**b**) indicates tumor region. **c, d** The relative proportion of SPOP (**c**) and ASCT2 (**d**) IHC staining in adjacent non-tumor and tumor tissues of breast. **e** The correlation analysis between SPOP and ASCT2 in breast tumor sections. **f** The association of SPOP and ASCT2 expression with overall survival in breast cancer patients. Survival fractions were plotted using the Kaplan–Meier method (Log-rank test). **g, h** Paired human breast normal and tumor tissues were analyzed by targeted metabolomics, and relative abundance of glutamine (**g**), glutamine/glutamate ratio (**h**) were compared between each individual pair of normal vs. tumor tissues. $n = 12$; Two-tailed, paired $t$ test. **i** A working model (created in Adobe Illustrator). Under glutamine-rich condition, SPOP forms dimer to promote ASCT2 ubiquitylation and degradation. Under glutamine-deprivation, GRK2 phosphorylates SPOP to inhibit its dimerization, which triggers SPOP self-ubiquitylation and degradation, leading to increased glutamine metabolism. Source data are provided as a Source Data file.

Desheng Lu (Shenzhen University, China). si-RNAs were synthesized by RiboBio (Guangzhou, China). The sequences of si-RNAs are as follows: ASCT2#1: 5′-GUACCGUCCUCAAUGUAGA-3′; ASCT2#2: 5′- GAAGCACAGAGCCUGA-GUU-3′; CK1δ#1: 5′-GGAGACAUCUA UCUCGGUA-3′; CK1δ#2: 5′- GCAAC-CUGGUGUACAUCAU-3′; CUL-1#1: 5′- GCUCUACACUCAUGUUUAU-3′; CUL-1#2: 5′- GAACCCAGUUACUGAAUAU -3′; CUL-2#1: 5′- GCCCUUAC-GUCAGUUGUAAAUUACA-3′; CUL-2#2: 5′- GA GCUAGCAUUGGAUAU-GUGG-3′; CUL-3#1: 5′- GCACAUGAAGACUAUAGU A-3′; CUL-3#2: 5′-GAGUGUAUGAGUUCCUAUU-3′; CUL-4A#1: 5′- GAAGC UGGUCAUCAA-GAAC-3′; CUL-4A#2: 5′- GAACUUCCGAGACAGACCU-3′; CUL-4B#1: 5′-AAGCCUAAAUUACCAGAAA-3′; CUL-4B#2: 5′- CACCGUCU CUAG-CUUUGCUAA-3′; CUL-5#1: 5′- GUCUCACUUCCUACUGAACUG-3′; CUL-5#2: 5′- CUGGAGGACUUGAUACCGGAA-3′; GRK2#1: 5′- AAGAAGUA CGAGAAGCUGGAG-3′; GRK2#2: 5′-GGCCAUGAGGAAGACUACGCC-3′; NAEβ#1, 5′-GCUUCUCUCUGCAAAUGAAAU-3; NAEβ#2, 5′-GCUACCAGAACA CUGUAUU-3′; SPOP#1: 5′- CAACUAUCAUGCUUCGGAU-3′; SPOP#2: 5′-GGUAAAGGUUCCUGAGUGC-3′; SPOP#3: 5′- AAAUGGUGUUUGCGAGUA A-3′; SPOP (3′-UTR): 5′- CUCCGUUUAAUUUCCAGAAUU-3′[38]. The sequences of shRNAs are as follows: shASCT2: CCGGCTGGATTATGAGGAATGGATACT CGAGTATCCATTCCTCATAATCCAGTTTTTG[74]; shSPOP: 5′- CAACUAU-CAU GCUUCGGAU-3′[38].

**Immunoblots and immunoprecipitation**. For direct immunoblot analysis, cells were lysed with RIPA lysis buffer, and the proteins were lysed from cells and quantified with the Bio-Rad protein assay kit (Bio-Rad laboratories, 500-0002EDU). The denatured proteins were subjected to SDS-PAGE, transferred to PVDF membranes, followed by immunoblot analysis with various antibodies.

For immunoprecipitation analysis, cells were lysed in an IP lysis buffer (50 mM Tris-HCl, pH 8.0, 120 mM NaCl, 0.5% NP40, 1 mM EDTA), supplemented with complete protease inhibitor cocktail (complete Mini, Roche). The lysates were incubated with bead-conjugated FLAG or HA antibody in a rotating incubator overnight at 4 °C. The immunoprecipitates were washed with lysis buffer and detected by immunoblot analysis.

**Protein half-life measurement**. Cells were transfected or treated under indicated conditions, followed by treatment with CHX (50 µg/ml) for various time points, and collected for immunoblotting. The band density was quantified using ImageJ software 1.48 v.

**The in vivo ubiquitylation assay**. HEK293 cells were transfected with various plasmids. Approximately 48 h later, the transfected cells were treated with MG132 (10 µM) for 6 h. The cells were then lysed in the guanidine denaturing solution (6 M guanidinium-HCl, 10 mM Tris HCl pH 8.0, 0.1 M $Na_2HPO_4$/$NaH_2PO_4$, 10 mM β-mercaptoethanol). After completely sonicated, the whole-cell lysates were incubated with Ni-NTA agarose (Qiagen, 1018244) for 4 h at room temperature (RT). Ni-NTA agarose was then successively washed once with each of denaturing solution, buffer A (8 M urea, 10 mM Tris HCl, pH 8.0, 0.1 M $Na_2HPO_4$/$NaH_2PO_4$, 10 mM β-mercaptoethanol), buffer B (8 M urea, 10 mM Tris HCl, pH 6.3, 0.1 M $Na_2HPO_4$/$NaH_2PO_4$, and 10 mM β-mercaptoethanol) with 0.2% Triton X-100, and buffer B with 0.1% Triton X-100 for 5 min in each step at RT. Ubiquitylated proteins were eluted from Ni-NTA agarose with the elution buffer (200 mM imidazole, 0.15 M Tris HCl, pH 6.7, 0.72 M β-mercaptoethanol, 5% SDS, 30% glycerol) for 20 min at RT and detected by immunoblotting using indicated antibodies.

**Quantitative real-time RT-PCR assay**. The qRT-PCR was performed as described previously[75]. Briefly, total cellular RNA was isolated using a TRIzol Reagent (Ambion, 15596018), and RNase-free DNase (Takara, 2270 A) was added to eliminate DNA contamination. The first strand cDNA was synthesized using a PrimeScript™ RT reagent Kit (Perfect Real Time) (Takara, RR037Q). About 100 ng cDNA from each sample was used to analyze gene expression with SYBR *Premix EX Taq* (Takara, RR420A). β-actin was used as an internal control. The PCR primers specific for each gene are listed below (5′-3′): *β-ACT*-F: TCACCCACACTGTGCCCATCTAC, *β-ACT*-R: GGAACCGGCTCATTGC CAATG; *ASCT2*-F: CATCGTCTTTGGTGTGGCCG, *ASCT2*-R: CACAGGGGCGTA CCACATG; *GLS*-F: GCTGTGCTCCATTGAAGTGA, *GL*-

*S*-R: GCAAACTGCCCT GAGAAGTC; *GLUD1*-F: TGCATGGCTTAACCTGGTGA T, *GLUD1*-R: TCTGGG CAGCTCACAATAAAGT; *GOT2*-F: TGAAGCCTTTAAG AGGGACACC, *GOT2*-R: CAAAATTCAGCCAGTCCCCC; *SNAT1*-F: GCAATGAC TCCAATGATTTCACC, *SNAT1*-R: ACTGCCCATAATGGCGTTG; *SNAT2*-F: CTT TTCATTTGTCTGTCATC CTGC, *SNAT2*-R: CACAGCCAGACGGACAATGAG.

**Cell viability and clonogenic survival assay**. Cells were harvested after various treatment, suspended in PBS and mixed with 0.4% solution of trypan blue (Sigma). Viable cells were counted. For clonogenic survival assay, cells (500 per dish) were seeded in 3.5-cm dishes, and cultured in medium containing various compounds for 10 days. The colonies were fixed, stained, and counted under an inverted microscope. Colonies with 50 cells or more were counted.

**Measurement of glutamine consumption and glutamate production**. Glutamine uptake/comsumption and glutamate production were measured with glutamine Assay Kit (Abnova, KA1627) and Glutamate Assay Kit (Sigma, MAK004-1KT) following the manufacturer's instructions. Glutamine uptake/consumption was calculated by deducting the measured glutamine concentration in the medium from the original glutamine concentration. All values were normalized according to cell number.

**The in vivo tumorigenesis assay**. Five- to six-week-old BALB/c athymic nude mice (nu/nu, female) were purchased from Shanghai SLAC Laboratory Animal Center. These mice were housed in the specific pathogen-free (SPF) environment at a constant temperature (25 °C) and a relatively constant humidity with ad libitum access to water and food (40–60%), with 12 h dark/light cycle. All animal studies were approved by and conducted in accordance with the guidelines established by the committee on Use and Care of Animals at the Zhejiang University.

MDA-MB-231 cells ($4 \times 10^6$) were injected subcutaneously. When the tumor size reached approximately 100 mm3, the mice were randomized. MLN4924 (30 mg/kg) or vehicle control was given to mice by subcutaneous injection, 5 days a week, and V-9302 (20 mg/kg) was given to mice via intraperitoneal injection, 5 days a week, respectively.

In a separate experiment, MDA-MB-231 cells ($4 \times 10^6$) expressing indicated shRNAs were injected subcutaneously. Tumor size and body weight were measured at the indicated time points and average tumor volumes were calculated according to the formula (length × width × width)/2. At the end of experiment, tumors were harvested and weighed.

**Immunohistochemistry (IHC) staining of human breast cancer tissues**. The study was approved by the ethical committee of Sir Run Run Shaw Hospital, Zhejiang University School of Medicine, China. In addition, informed consent was obtained from all of the subjects involved. Human breast tumor tissue arrays were obtained from Sir Run Run Shaw Hospital. 314 female cases were recruited with ages ranging from 24 to 75. The IHC analysis staining of paraffin-embedded tissue sections was carried out using the Dako Envision system (Dako, Glostrup, Denmark) as previously described[76]. Briefly, the sections were submerged in boiling 10 mmol/L sodium citrate (pH, 6.0) for 2 min in a pressure cooker. After being treated with 0.3% hydrogen peroxide for 10 min to block endogenous peroxidase, the sections were incubated with primary antibody overnight at 4 °C. After washing, the sections were incubated with biotin-labeled secondary immunoglobulin (Dako) for 40 min at room temperature, followed by incubation with 3, 3′-diaminobenzidine (Dako) at room temperature. Images were taken by using an Olympus camera and matched software. The following primary antibodies were used: rabbit-ASCT2 (Abcam, ab84903; diluted at 1:800), rabbit-SPOP (Affinit, DF12106; diluted at 1:800).

**Metabolic profiling analysis**. The untargeted metabolic profiling of cell extracts and culture supernatants were performed and analyzed by Metabo-Profile (Shanghai, China) as described in our previous study[31]. Metabolites with variable importance in projection (VIP) of ≥1 and $p < 0.05$ were regarded as statistically significant (differentially expressed metabolites). Pathway analysis was performed by integrating hypergeometric test and topology analysis. The pathway impact value was calculated by an Impact factor (IF) analytic tool, as defined previously[77] to measure the degree of involvement of various metabolic pathways.

**Sample preparation for metabolomics**. The study was approved by the ethical committee of Sir Run Run Shaw Hospital, Zhejiang University School of Medicine, China. In addition, informed consent was obtained from all of the subjects involved. 12 female cases with breast cancer were recruited with ages ranging from 38 to 74. Each paired human breast tissue samples (tumor vs. normal) were weighed and homogenized in ice-cold 80% methanol aqueous solution (50 mg tissue/mL). Then, 100 μL of tissue homogenate was mixed with 900 μL ice-cold 80% methanol/water. The mixture was vortexed and centrifuged at 21,130×g for 15 min at 4 °C. The supernatant (~900 μL) was transferred to a new tube, whereas the pellet was mixed with 500 μL ice-cold 80% methanol/water and re-centrifuged at $21,130 \times g$ for 15 min at 4 °C. The supernatant (~500 μL) was collected and combined with the previous supernatant. The mixed supernatant was vortexed and divided into two new tubes with equal volume. The supernatant was dried with SpeedVac (Thermo Scientific) and the dried metabolite extracts were stored at −80 °C until LC-MS/MS-based metabolomics analysis.

The samples for quality control (QC) were prepared by pooling equal aliquot of homogenate from each tissue sample. Pretreatment of the QC samples was in parallel with that of the study samples.

**Targeted metabolomics analysis**. Dried metabolites were reconstituted in LC-MS grade water with 0.03% formic acid, vortex-mixed, and centrifuged at 4 °C for 15 min to remove debris. Samples were randomized and blinded before analyzing by LC-MS/MS. Chromatographic separation was performed on a Nexera UHPLC system (Shimadzu), with a RP-UPLC column (HSS T3, 2.1 mm × 150 mm, 1.8 μm, Waters) and the following gradient: 0–3 min 99% A; 3–15 min 99–1% A; 15–17 min 1% A; 17–17.1 min 1–99% A; 17.1–20 min 99% A. Mobile phase A was 0.03% formic acid in water. Mobile phase B was 0.03% formic acid in acetonitrile. The flow rate was 0.25 mL/min, the column was at 35 °C and the autosampler was at 4 °C. Mass data acquisition was performed using an AB QTRAP 6500+ triple quadrupole mass spectrometer (SCIEX, Framingham, MA) in multiple reactions monitoring (MRM) mode, as previously described with minor modifications[78]. Chromatogram review and peak area integration were performed using Multi-Quant 3.0.2 (SCIEX, Framingham, MA).

**Metabolomics data processing**. QC samples were inserted in an interval of every ten test samples to monitor the stability of instrument and normalize the variations during the run. Briefly, the mean peak area of each metabolite from all the QC samples in all given batches (QCall), as well as the mean peak area of each metabolite from the QC samples that are the most adjacent to a given group of test samples (QCadj) were first calculated. The ratio between these two mean peak areas for each metabolite was computed by dividing the same QCall by each QCadj and used as the normalization factor for each given group of test samples. The peak area of each metabolite from each test sample was normalized by multiplying its corresponding normalization ratio to obtain the normalized peak area. In addition, to effectively correct the sample to sample variations in biomass that may contribute to systematic differences in metabolite abundance detected by LC-MS, we generated the scaled data by comparing the normalized peak area of each metabolite to the sum of the normalized peak area from all the detected metabolites in that sample.

**Statistical analyses**. All statistical analysis were performed using the GraphPad Prism version 8.0.2.263 and SPSS 16.0. All statistical comparisons were evaluated by the Student's $t$-test or one-way or two-way analysis of variance (ANOVA). Among all the data sets, $p$ values less than 0.05 were considered significant.

Immunoblot assay were performed at least two independent times, and protein half-life measurement were performed with three independent repeats to confirm reproducibility, respectively.

**Reporting summary**. Further information on research design is available in the Nature Research Reporting Summary linked to this article.

## Data availability

All experimental data that support the findings of this study are included as follows: 7 main figures, 9 Supplementary figures, and Source data. Full metabolomics data can be found in the Source Data file. All the data are available from the corresponding author, Yi Sun. Source data are provided with this paper.

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

## Acknowledgements

We would like to thank Drs. Ping Wang, Xinbo Wang, and Desheng Du for providing various expressing plasmids. We would also like to thank the staff members from the Core facilities and the Morphological Platform at Zhejiang University School of Medicine for their assistance in data collection and technical support.

This work is supported in part by the National Key R&D Program of China (2021YFA1101000 and 2016YFA0501800 to YS), Chinese NSFC grant (31701167 to QZ), and Zhejiang Provincial Natural Science Foundation of China (LD22H300003 to YS).

## Author contributions

Q.Z., H.J., X.W. and Y.S. conceived and designed the research, and Q.Z. performed most of the experiments; WL helped in animal experiments, and quantitative real-time RT-PCR assay; C.W. performed the immunohistochemistry assays and helped relevant data analysis; F.S., Y.W. performed the targeted metabolomics and analyzed the data under the guidance of Z.H.; S.J., Y.C. and L.W. collected and provided part of human breast cancer samples; Q.C. performed part of protein half-life experiments and helped Q.Z. finalize the revised manuscript; H.L. performed bioinformatics analysis; Q.Z., H.J., X.W., and Y.S. analyzed and interpreted the data; Q.Z. and Y.S. wrote the manuscript. Y.S. oversaw the project and finalized the manuscript. The manuscript was approved by all authors.

## Competing interests

The authors declare no competing interests.
