## [Peer Review File · Nature Communications]

Neddylation inhibition induces glutamine uptake and metabolism by targeting CRL3^{SPOP} E3 ligase in cancer cellsREVIEWER COMMENTS

Reviewer #1 (Remarks to the Author); expert on phosphorylation, ubiquitination:

In the present study the authors first examined how inhibition of neddylation affects changes in global metabolism and found that treatment of neddylation inhibitor MLN4924 significantly affected glutamine metabolism. By surveying the effect on several primary transporters, they narrowed down ASCT2/SLC1A5 is the major target of MLN compound. They identified CUL3/SPOP is the E3 ubiquitin ligase complex mediating degradation of ASCT2. Additionally, they showed that SPOP binds to a consensus SBC motif in ASCT2 and phosphorylation of this motif by CK1 further facilitates degradation of ASCT2 by SPOP. Furthermore, they showed that SPOP can be self-ubiquitinated and this process is regulated by SPOP phosphorylation by GRK2 under glutamine deprivation conditions. Finally, they showed that expression of SPOP and ASCT2 is inversely correlated in breast cancer patient samples and that SPOP ASCT2 stabilization and increased glutamine uptake due to SPOP downregulation are important for tumor growth in vitro and in vivo.

Overall, the findings are very novel, especially expand our understanding the role of SPOP in regulating cellular metabolism and tumor growth. Also, findings are highly mechanistic and clinically relevant. However, a number of issues need to be clarified and the quality of a few experiments need to be improved for publication in Nature Communications.

Major points:

The authors miscited a few references in terms of previous studies related to SPOP regulation of cellular metabolisms. For example, it was An and colleagues (Cell Reports 2014), but not Geng et al. who are the first to report that androgen receptor is the bona fide degradation target of SPOP. Also, it was Zhang and colleagues (Nature Medicine 2017) who are the group reporting the role of SPOP in regulating the stability of the key transcription regulators BET proteins and the downstream cholesterol metabolism pathways. Plus, Shi and colleagues have shown recently that SPOP regulates testosterone synthesis metabolism (Cancer Research 2021). All these relevant studies should be cited appropriately and unrelated references should be avoided.

In Figure 1a and 1b, it is unclear how the Pathway Impact is calculated and that the scores mean in the X axis. The detailed information should be provided in either figure legends or the Methods section.

There are so many bands in the WB of ASCT2 throughout the study. Are they different isoforms or PTMs or simply some bands are nonspecific. The issue needs to be addressed.

Knockdown of CUL3 had the most drastic effect on the increase in ASCT2 protein levels. It appears that CUL2 KD also had similar trend although it is not drastic as CUL3 KD. The authors should either acknowledge this potential mechanism or rule out this possibility by repeating the experiments in multiple cell lines.

The authors indicated that MLN treatment had minimal effect on the mRNA levels of a few primary transporters. However, whether the differences in groups with or without MLN treatment are statistically significant or not is untested. If the differences are statistically significant, the authors then should at least need to discuss the other possible regulations of these transporter genes by the CUL3/CUL1 pathways.

To unbiasedly determine which ubiquitin linkage is involved in poly-ubiquitination of ASCT2 by SPOP, all seven lysine residues in Ub should be mutated individually and tested.

The authors should make it clear that phosphorylation of a substrate on the serine residues of the SPOP binding motif is NOT prerequisite for its binding with SPOP and subsequent ubiquitination and degradation per se. It is true that this has been reported occasionally, but it is not common. While SRC-3 is a well established SPOP target, it does not harbor a perfectly matched SPOP bound SBC

motif. It seems this ‘shortcoming’ can be perfectly compensated by phosphorylation of SRC-3 by CK1. Although ERG protein has a perfectly matched SPOP bound SBC motif, it has been implicated that approximately 10 more amino acids immediate adjacent to the ERG SBC also play an important role in influencing the binding of ERG by SPOP (An et al, Mol Cell 2015). Therefore, in the case of ERG, it is not surprising that phosphorylation of SBC by CK1 can fine tune the binding and regulation of ERG by SPOP. In case of ASCT2, it could be the case like ERG since authors’ data clearly imply that other CUL proteins can also regulate its degradation. Nevertheless, the rationale for the authors to study the role of CK1 in regulation of SPOP-ASCT2 and ASCT2 ubiquitination should be rephrased.

The data shown in Figure 4A is convincing, but the author’s interpretation appears off the wheel. Instead of indicating that the effect of 3A mutant is caused by blocking phosphorylation of SBC motif, it is also possible that this mutant impaired the function of the SBC motif itself. To prove their hypothesis is true, however, they could use phospho-mimetic mutants such as 3D or 3E mutants, rather than 3A mutant.

In order to fully support the hypothetical model the authors proposed, the authors should clarify whether it is known in literature or they have the evidence showing that glutamine deprivation leads to the activation of GRK2.

The working model depicted in Figure 7K is nice, but can be improved. It is hard to recognize that the “bigger” shape of CUL3 and SPOP in ‘normal’ tissues represents more proteins. It would be more straightforward to include SPOP molecules in “normal” tissues. Additionally, the authors only showed the data that SPOP is downregulated in tumor tissues, but not CUL3. Therefore, it seems appropriate to remove CUL3 in this model.

Reviewer #2 (Remarks to the Author); expert on metabolism, breast cancer:

In this manuscript, Zhou and colleagues present evidence that ASCT2, and therefore glutamine metabolism, are altered by Cul3-SPOP. ASCT2 is certainly an important molecule in breast cancer and mechanisms regulating ASCT2 are of interest to the cancer and metabolism communities. In many cases the experiments seem to be performed well, and the authors provide a lot of mechanistic understanding of the proposed pathways. However, it seems that the authors may have extended knowledge of the identified regulatory pathways at the expense of more detailed and thorough characterization of the most important and proximal components of their study. Examples of this and other issues with the manuscript are highlighted below along with several suggestions that may strengthen the conclusions drawn by the authors.

The authors present a lot of immunoprecipitation data showing various interactions between SPOP, ASCT2, CK1d, GRK, etc. However, the vast majority of these interactions are observed on overexpressed HA- or FLAG-tagged proteins, and dramatically overexpressed proteins can sometimes interact with proteins in ways that are not seen under normal expression levels. The authors should include IP experiments using non-overexpressed proteins to provide confidence that the interactions they have identified are physiologically relevant.

For both the identification of CK1d as a kinase for ASCT2 and GRK2 as a kinase for SPOP, the authors use inhibitors and overexpression of mutant proteins to suggest that these kinases are regulating the expression/activity of these proteins. While the presented data supports that these kinases are involved in the regulation of these proteins, the authors need to show that the substrates are actually being phosphorylated by the identified kinases.

On line 273 the authors conclude that glutamine deprivation induces SPOP self-ubiquitylation. Given that these experiments were performed in cells, how do the authors know that this is self-ubiquitylation and not ubiquitylation by some other E3 ligase? There is no data specifically supporting

the self-ubiquitylation model.

For the in vivo experiments presented in Figure 6, the authors do not show the level of knockdown of ASCT2 and SPOP. The authors only utilize one shRNA in this experiment, as they also do for most of the siRNA experiments. Individual si and shRNAs often have off-target effects and use of 2 separate si/shRNAs is preferred.

On lines 171-173, the authors conclude that "MLM4924 enhanced glutamine metabolism via increasing the levels of glutamine transporter ASCT2..." While the data shown in Figure 1 does support this conclusion, it would be necessary to show that overexpression of ASCT2 is sufficient to induce glutamine consumption in these cells. This type of data is eventually shown in Figure 6, but the authors only show a blot for HA to show ASCT2 overexpression, which does not allow readers to determine whether the ASCT2 is expressed at similar levels to that seen after treatment with MLN. The authors should be careful with their conclusion as the manuscript progresses and be sure to show the relevant controls in order to support their conclusions.

In figure 7 the authors present data that glutamine metabolism is altered in breast cancer, which has been shown numerous times by other reports. The authors also show GSSG and cystine data, but it is unclear what the real point of this data is given that they never show GSSG or cystine through the rest of the manuscript. As it stands, the analysis of glutamine metabolism in Figure 7 seems very superficial and does not add much to the manuscript.

ASCT2 has numerous substrates in addition to glutamine, but the authors do not mention any of the other potential implications that their work could have on other amino acid metabolic pathways.

The inhibitor V9302 has been shown to be nonspecific to ASCT2 (Bröer A, Fairweather S and Bröer S (2018) Disruption of Amino Acid Homeostasis by Novel ASCT2 Inhibitors Involves Multiple Targets. *Front. Pharmacol.* 9:785). As such, it is not clear how the use of V9302 supports the authors' conclusions.

Reviewer #3 (Remarks to the Author); expert on neddylation, metabolism:

The manuscript entitled "Regulation of glutamine uptake and metabolism by the SPOP-ASCT2 axis" describes the regulation of the E3 ligase SPOP and the glutamine transporter ASCT2 by the neddylation inhibitor MLN4924. They also describe the association between glutamine levels and SPOP and ASCT2 in breast cancer tumors.

Although this is a detailed study from a mechanistic point of view, the authors do not relate their findings to NEDD8 levels or the neddylation cycle. In this regard my comments are as follows:

1. after blocking neddylation with MLN4924 how is mitochondrial activity modulated? measurements of OCR, mitochondrial ROS and ATP would be important to understand the metabolic rewiring.
 2. after MLN4924 treatment, is the response in SPOP and ASCT2 levels dependent or independent of glutamine levels?
 3. Are SPOP and ASCT2 levels related to the levels of NEDD8 and/or the enzymes of the neddylation cycle?
 4. are glucose levels modulators of SPOP and ASCT2. Does the effect shown exhibit greater dependence on glucose or glutamine?
 5. What is the effect of MLN4924 on CK1 δ , does it exert any regulation?
 6. What is the in vivo correlation between SPOP and ASCT2, and the levels of the enzymes involved in neddylation?
 7. What happens with the levels of NEDD8 and the enzymes involved in the neddylation cycle with the levels of glutamine?
- Could SPOP be a neddylation target?
9. Do glutamine levels regulate SPOP neddylation?

Re: MS ID: NCOMMS-21-31948

MS TITLE: Regulation of glutamine uptake and metabolism by the SPOP-ASCT2 axis

RESPONSES TO REVIEWER COMMENTS

Reviewer #1 (Remarks to the Author); expert on phosphorylation, ubiquitination:

In the present study the authors first examined how inhibition of neddylation affects changes in global metabolism and found that treatment of neddylation inhibitor MLN4924 significantly affected glutamine metabolism. By surveying the effect on several primary transporters, they narrowed down ASCT2/SLC1A5 is the major target of MLN compound. They identified CUL3/SPOP is the E3 ubiquitin ligase complex mediating degradation of ASCT2. Additionally, they showed that SPOP binds to a consensus SBC motif in ASCT2 and phosphorylation of this motif by CK1 further facilitates degradation of ASCT2 by SPOP. Furthermore, they showed that SPOP can be self-ubiquitinated and this process is regulated by SPOP phosphorylation by GRK2 under glutamine deprivation conditions. Finally, they showed that expression of SPOP and ASCT2 is inversely correlated in breast cancer patient samples and that SPOP ASCT2 stabilization and increased glutamine uptake due to SPOP downregulation are important for tumor growth in vitro and in vivo.

We thank this reviewer for his/her excellent summary of our work.

Overall, the findings are very novel, especially expand our understanding the role of SPOP in regulating cellular metabolism and tumor growth. Also, findings are highly mechanistic and clinically relevant. However, a number of issues need to be clarified and the quality of a few experiments need to be improved for publication in Nature Communications.

We thank this reviewer for his/her acknowledgement of the novelty and significance of our finding. We also thank the reviewer for the constructive comments.

1) Major points:

The authors miscited a few references in terms of previous studies related to SPOP regulation of cellular metabolisms. For example, it was An and colleagues (Cell Reports 2014), but not Geng et al. who are the first to report that androgen receptor is the bona fide degradation target of SPOP. Also, it was Zhang and colleagues (Nature Medicine 2017) who are the group reporting the role of SPOP in regulating the stability of the key transcription regulators BET proteins and the downstream cholesterol metabolism pathways. Plus, Shi and colleagues have shown recently that SPOP regulates testosterone synthesis metabolism (Cancer Research 2021). All these relevant studies should be cited appropriately and unrelated references should be avoided.

We thank the reviewer for raising this important issue. Per review's suggestion, we have now cited these important and insightful studies to replace our previously mis-cited ones.

2) In Figure 1a and 1b, it is unclear how the Pathway Impact is calculated and that the

scores mean in the X axis. The detailed information should be provided in either figure legends or the Methods section.

The point is well-taken. We have now added the detailed information in the Methods section (pages 26-27).

3) There are so many bands in the WB of ASCT2 throughout the study. Are they different isoforms or PTMs or simply some bands are nonspecific. The issue needs to be addressed.

We thank the reviewer for this important concern. It has been reported previously that ASCT2 contains several N-linked glycosylation sites and as the results, displayed several bands in western blotting (Console et al., 2015). We have now experimentally addressed the reviewer's concern. N-glycosidase F (PNGase F) treatment to remove the glycosylation of cell lysates from several breast and lung cancer cell lines converted multiple bands (with molecular mass varying from 55-100KDa) of ASCT2 into one unmodified band with expected size. Thus, the multiple bands of ASCT2 we detected are indeed due to ASCT2 glycosylations, not nonspecific ones. We have included the data as Supplementary Fig. 1i, and described it in the text (pages 6-7).

4) Knockdown of CUL3 had the most drastic effect on the increase in ASCT2 protein levels. It appears that CUL2 KD also had similar trend although it is not as drastic as CUL3 KD. The authors should either acknowledge this potential mechanism or rule out this possibility by repeating the experiments in multiple cell lines.

Thank you for a valid point. To determine the possible involvement of Cullin-2 in regulation of ASCT2 levels, we repeated this experiment in two breast cancer cell lines MDA-MB-231 and BT549, and found a minimal, if any, effect on ASCT2 levels upon Cul-2 KD (Supplementary Fig. 2a, b). Thus, Cullin-3 is our major focus in this study.

5) The authors indicated that MLN treatment had minimal effect on the mRNA levels of a few primary transporters. However, whether the differences in groups with or without MLN treatment are statistically significant or not is untested. If the differences are statistically significant, the authors then should at least need to discuss the other possible regulations of these transporter genes by the CUL3/CUL1 pathways.

Good point. We have now determined potential effect of MLN4924 on the mRNA expression of few primary transporter genes. Although ASCT2, SNAT1, SNAT2, GLS, and GOT2 gene expression in BT549 cell line were statistically increased after MLN4924 treatment, but the degree was much lesser than the changes in protein levels. Given primary effect of MLN4924 is on blocking cullin neddylation to inactivate cullin-RING ligases, its effect on mRNA level is likely indirect. Per reviewer's suggestion, we have now mentioned this point in the Results section (page 6), and discussed it in the Discussion section (pages 17-18).

6) To unbiasedly determine which ubiquitin linkage is involved in poly-ubiquitination of

ASCT2 by SPOP, all seven lysine residues in Ub should be mutated individually and tested.

We performed the experiments, as suggested. We demonstrated that polyubiquitylation of ASCT2 by SPOP is mainly via the K48-linkage (newly generated Supplementary Fig. 3g).

7) The authors should make it clear that phosphorylation of a substrate on the serine residues of the SPOP binding motif is NOT prerequisite for its binding with SPOP and subsequent ubiquitination and degradation per se. It is true that this has been reported occasionally, but it is not common. While SRC-3 is a well-established SPOP target, it does not harbor a perfectly matched SPOP bound SBC motif. It seems this ‘shortcoming’ can be perfectly compensated by phosphorylation of SRC-3 by CK1. Although ERG protein has a perfectly matched SPOP bound SBC motif, it has been implicated that approximately 10 more amino acids immediate adjacent to the ERG SBC also play an important role in influencing the binding of ERG by SPOP (An et al, Mol Cell 2015). Therefore, in the case of ERG, it is not surprising that phosphorylation of SBC by CK1 can fine tune the binding and regulation of ERG by SPOP. In case of ASCT2, it could be the case like ERG since authors’ data clearly imply that other CUL proteins can also regulate its degradation. Nevertheless, the rationale for the authors to study the role of CK1 in regulation of SPOP-ASCT2 and ASCT2 ubiquitination should be rephrased.

The point well-taken and thank you. We have now rephrased our statement in the Results section and cited this publication (page 9).

8) The data shown in Figure 4A is convincing, but the author’s interpretation appears off the wheel. Instead of indicating that the effect of 3A mutant is caused by blocking phosphorylation of SBC motif, it is also possible that this mutant impaired the function of the SBC motif itself. To prove their hypothesis is true, however, they could use phospho-mimetic mutants such as 3D or 3E mutants, rather than 3A mutant.

Agreed. We have now included a phosphor-mimetic mutant ASCT2-3D in the assay, and found that this mutant has an enhanced binding with SPOP, while -3A mutant has a reduced binding, further supporting the notion that phosphorylation on SBC motif indeed facilitates the SPOP binding. This newly generated data is now included in Fig.4a and described in the text (page 9).

9) In order to fully support the hypothetic model the authors proposed, the authors should clarify whether it is known in literature or they have the evidence showing that glutamine deprivation leads to the activation of GRK2.

Again, a valid point. We searched the literatures and found a paper reporting that glutamine attenuates itch via a rapid induction of GRK2 in mouse ear skin tissues (Im et al., 2018). Interestingly, our research found that glutamine deprivation leads to the activation of GRK2 in breast cancer cell lines. Thus, a possible explanation is that glutamine has different roles in the regulation of GRK2 in a context (tissue/cell)

dependent manner. We have now included this discussion in the Discussion section in the revised manuscript (pages 18-19).

10) The working model depicted in Figure 7K is nice, but can be improved. It is hard to recognize that the “bigger” shape of CUL3 and SPOP in ‘normal’ tissues represents more proteins. It would be more straightforward to include SPOP molecules in “normal” tissues. Additionally, the authors only showed the data that SPOP is downregulated in tumor tissues, but not CUL3. Therefore, it seems appropriate to remove CUL3 in this model.

Yes, we modified the model per reviewer’s suggestion, but kept Cul-3 in the figure, given SPOP has to complex with Cul-3 to function as an active E3 (Figure 7i).

Reviewer #2 (Remarks to the Author); expert on metabolism, breast cancer:

In this manuscript, Zhou and colleagues present evidence that ASCT2, and therefore glutamine metabolism, are altered by Cul3-SPOP. ASCT2 is certainly an important molecule in breast cancer and mechanisms regulating ASCT2 are of interest to the cancer and metabolism communities. In many cases the experiments seem to be performed well, and the authors provide a lot of mechanistic understanding of the proposed pathways. However, it seems that the authors may have extended knowledge of the identified regulatory pathways at the expense of more detailed and thorough characterization of the most important and proximal components of their study. Examples of this and other issues with the manuscript are highlighted below along with several suggestions that may strengthen the conclusions drawn by the authors.

We thank the reviewer for his/her positive comments and appreciation of our work. We have performed new experiments to address the comments raised in details below.

1) The authors present a lot of immunoprecipitation data showing various interactions between SPOP, ASCT2, CK1d, GRK, etc. However, the vast majority of these interactions are observed on overexpressed HA- or FLAG-tagged proteins, and dramatically overexpressed proteins can sometimes interact with proteins in ways that are not seen under normal expression levels. The authors should include IP experiments using non-overexpressed proteins to provide confidence that the interactions they have identified are physiologically relevant.

We thank the reviewer for these valid suggestions. We have now performed these IPs under normal cultured condition without overexpression. These newly generated data on the interaction between SPOP and ASCT2 (Fig. 2d), ASCT2 and CK1 δ (Fig. 4c), and SPOP and GRK2 (Fig. 5j, Supplementary Fig. 6h) were included in this revision to show that these bindings are physiologically relevant, and described in the text.

2) For both the identification of CK1d as a kinase for ASCT2 and GRK2 as a kinase for SPOP, the authors use inhibitors and overexpression of mutant proteins to suggest that these kinases are regulating the expression/activity of these proteins. While the presented data supports that these kinases are involved in the regulation of these proteins, the

authors need to show that the substrates are actually being phosphorylated by the identified kinases.

We thank the reviewer for raising this excellent point. To address it, we generated phosphor-S222-SPOP Ab and found that SPOP was indeed phosphorylated on Ser222 in response to glutamine withdrawal (Fig. 5m, Supplementary Fig. 6l). With regards to generating ASCT2 phosphor-Ab, we decided not to pursue it, based upon the following two reasons: 1) As reviewer 1 pointed out (point #7), phosphorylation on SPOP substrate (such as ASCT2 in our study) is not a prerequisite event for SPOP mediated ubiquitylation and degradation; 2) It will be technically challenging to raise a highly specific phosphor-Ab against a peptide with all three serine residues phosphorylated. We hope that the reviewer will agree with us. Thank you.

3) On line 273 the authors conclude that glutamine deprivation induces SPOP self-ubiquitylation. Given that these experiments were performed in cells, how do the authors know that this is self-ubiquitylation and not ubiquitylation by some other E3 ligase? There is no data specifically supporting the self-ubiquitylation model.

An excellent point, which is well taken. It is well known that the BTB domain of SPOP binds to Cullin 3 to form a complex with E3 ligase activity (Zhuang et al., 2009). The SPOP Δ BTB mutant would not bind to cullin 3, thus having no ligase activity. We hypothesized that if the ligase-dead SPOP Δ BTB mutant also undergoes polyubiquitylation after glutamine deprivation, it must be mediated by another E3, not by self-ubiquitylation. As shown in newly generated data (Supplementary Fig. 6e), wild-type SPOP is undergoing polyubiquitylation upon glutamine deprivation, whereas this SPOP Δ BTB mutant appears to be polyubiquitylated slightly, if any, by an unknown E3/mechanism. It is, however, completely independent of glutamine deprivation. We, therefore, concluded that glutamine deprivation indeed triggers SPOP self-ubiquitylation.

4) For the in vivo experiments presented in Figure 6, the authors do not show the level of knockdown of ASCT2 and SPOP. The authors only utilize one shRNA in this experiment, as they also do for most of the siRNA experiments. Individual si and shRNAs often have off-target effects and use of 2 separate si/shRNAs is preferred.

We thank the reviewer for these critical comments. As suggested, we performed the experiment and showed that the levels of ASCT2 and SPOP were indeed knocked down in tumor tissues. These newly generated data are now included as Fig. 6h. Moreover, we now performed all the experiments involving siRNA-based knockdown using two independent siRNAs, and the newly generated data are now incorporated into corresponding figures. For shRNAs targeting ASCT2 and SPOP, we referred the well-confirmed sequences from the published papers (Wang et al., 2015; Wang et al., 2019). We also confirmed the knockdown efficiency as shown in Fig. 6h.

5) On lines 171-173, the authors conclude that “MLM4924 enhanced glutamine metabolism via increasing the levels of glutamine transporter ASCT2...” While the data shown in Figure 1 does support this conclusion, it would be necessary to show that

overexpression of ASCT2 is sufficient to induce glutamine consumption in these cells. This type of data is eventually shown in Figure 6, but the authors only show a blot for HA to show ASCT2 overexpression, which does not allow readers to determine whether the ASCT2 is expressed at similar levels to that seen after treatment with MLN. The authors should be careful with their conclusion as the manuscript progresses and be sure to show the relevant controls in order to support their conclusions.

Good points! Per reviewer's suggestion, we directly compared ASCT2 levels upon ectopic expression vs. MLN4924 treatment, using Ab against ASCT2. We found that ASCT2 levels are higher under overexpression condition than that induced by MLN4924 treatment (Supplementary Fig. 7f). The observation that MLN4924 is more efficient in promoting glutamine uptake and glutamate production than that of HA-ASCT2 overexpression (Fig. 1c, d, Supplementary Fig. 1d, e vs. Fig. 6d, e and Supplementary Fig. 7d, e), suggests that MLN4924 may also has ASCT2-independent effect in modulating glutamine metabolism. We have now discussed this point in the text (page 13).

6) In figure 7 the authors present data that glutamine metabolism is altered in breast cancer, which has been shown numerous times by other reports. The authors also show GSSG and cystine data, but it is unclear what the real point of this data is given that they never show GSSG or cystine through the rest of the manuscript. As it stands, the analysis of glutamine metabolism in Figure 7 seems very superficial and does not add much to the manuscript.

The reviewer was correct that the others have previously shown that glutamine metabolism is altered in breast cancer. The power of our study is that we did a direct paired comparison using tumor vs. non-tumor tissues from the same patient. We agreed with the reviewer that GSSG or cysteine was never mentioned in the rest of manuscript, we therefore deleted these data, since they are indirectly related to glutamine metabolism.

7) ASCT2 has numerous substrates in addition to glutamine, but the authors do not mention any of the other potential implications that their work could have on other amino acid metabolic pathways.

Excellent point. Indeed, ASCT2 (Alanine, Serine, Cysteine Transporter -2), as name referred, is a transporter for neutral amino acids including alanine, Serine, cysteine, threonine, in addition to glutamine (Bhutia and Ganapathy, 2016; Scalise et al., 2017). Per reviewer's suggestion, we have added few sentences in the Discussion section for potential effect/implication of MLN4924 on other amino acid metabolic pathways (page 20).

8) The inhibitor V9302 has been shown to be nonspecific to ASCT2 (Bröer A, Fairweather S and Bröer S (2018) Disruption of Amino Acid Homeostasis by Novel ASCT2 Inhibitors Involves Multiple Targets. *Front. Pharmacol.* 9:785). As such, it is not clear how the use of V9302 supports the authors' conclusions.

A valid point. Indeed, the specificity is always a concern for any small molecule inhibitors. Given the fact that ASCT2 is an emerging therapeutic target for human cancers, a large number of ASCT2 inhibitors have been reported, including the widely used (in preclinical models) GPNA and V-9302 (Jiang et al., 2020; Wahi and Holst, 2019). V-9302 has been shown to block glutamine transport and inhibit cancer cell growth both in *vitro* and in *vivo* studies (Schulte et al., 2018; Zhang et al., 2020). So far, V-9302 is the best available ASCT2 inhibitors (Jiang et al., 2020), although some off-targeted effect was reported as reviewer mentioned in an oocyte expression model (Broer et al., 2018).

Our study used a combinational strategy, including genetic approach with siRNA-based ASCT2 knockdown and pharmacological approach with the use of both V-9302 and GPNA (Schulte et al., 2018), two reported small molecule inhibitors of ASCT2 to demonstrate that targeting ASCT2 indeed suppressed the growth of breast cancer cells in both *in vitro* cell culture and *in vivo* xenograft tumor models. Nevertheless, we cited paper by Broer et al. to acknowledge the potential off-target effect of V-9302 in the Discussion section (page 20).

Reviewer #3 (Remarks to the Author); expert on neddylation, metabolism:

The manuscript entitled "Regulation of glutamine uptake and metabolism by the SPOP-ASCT2 axis" describes the regulation of the E3 ligase SPOP and the glutamine transporter ASCT2 by the neddylation inhibitor MLN4924. They also describe the association between glutamine levels and SPOP and ASCT2 in breast cancer tumors. Although this is a detailed study from a mechanistic point of view, the authors do not relate their findings to NEDD8 levels or the neddylation cycle. In this regard my comments are as follows:

We thank the reviewer for his/her positive comment on the mechanistic nature of our study, and will address the concern on NEDD8 levels or the neddylation cycle.

1) after blocking neddylation with MLN4924 how is mitochondrial activity modulated? measurements of OCR, mitochondrial ROS and ATP would be important to understand the metabolic rewiring.

Excellent point. As a matter of fact, we have performed these experiments and reported the finding in 2019 (Zhou et al., 2019) (PMID: 30668548). In brief, MLN4924 induced mitochondrial fission-to-fusion conversion morphologically, and remarkably stimulated basal OCR, ATP-linked OCR, proton-leak OCR, maximal OCR, and non-mitochondrial OCR, while significantly suppressed reserve-capacity OCR. Moreover, MLN4924 significantly decreased ATP production and caused an increase in mito-ROS. We concluded that MLN4924 effectively impairs mitochondrial integrity, respiration, and functions. The current study was actually a continuation of this previous finding.

2) after MLN4924 treatment, is the response in SPOP and ASCT2 levels dependent or independent of glutamine levels?

Excellent point. We performed this suggested experiment by treating two lines of breast cancer cells with MLN4924 (different concentration) in glutamine-rich or glutamine-free media. The newly generated data (Fig. R1) showed that “after MLN4924 treatment, the response in SPOP and ASCT2 levels is dependent of glutamine levels”. Specifically, in glutamine-rich condition, MLN4924 treatment increased the levels of both ASCT2 and SPOP. However, under glutamine deprivation condition, MLN4924 treatment decreased ASCT2 levels (seen in MDA-MB-231), but increased SPOP levels (in both lines), while glutamine deprivation alone did just opposite (in both lines). Possible explanation is as follows: 1) as a general inhibitor of cullin-RING ligase, MLN4924 inhibited the self-ubiquitylation of SPOP induced by glutamine deprivation, leading to an increase; 2) Given CRL3-SPOP ligase activity is also inhibited by MLN4924, which prevented SPOP induced ubiquitylation and degradation of ASCT2, it is most likely that MLN4924-induced ASCT2 reduction seen in MDA-MB-231 cells is via an off-target effect independent of its neddylation inhibitory activity. This is an interesting topic for further investigation, but is out of scope of this study. We, therefore, decided not to include this piece of data in the manuscript.

Fig. R1: MDA-MB-231 (a) and BT549 (b) cells were treated with various concentrations MLN4924 and cultured with glutamine-rich or glutamine-free medium, respectively for 24 h, followed by Western blotting using indicated antibodies.

3) Are SPOP and ASCT2 levels related to the levels of NEDD8 and/or the enzymes of the neddylation cycle?

Very interesting comment. Per reviewer’s suggestion, we exogenously expressed NEDD8, and two neddylation E2s, UBE2M, and UBE2F, respectively and then determined their effect on the levels of SPOP and ASCT2. We found that overexpression of NEDD8, UBE2M, and UBE2F has minimal, if any, effects on the levels of SPOP and ASCT2. This newly generated data is now included in Supplementary Fig. 5j, k, and described in the text (page 11)

4) are glucose levels modulators of SPOP and ASCT2. Does the effect shown exhibit greater dependence on glucose or glutamine?

Another interesting comment. We now have done a head-to-head comparison of the effect on SPOP and ASCT2 by the withdrawal of glutamine vs. glucose. The result

showed that while glutamine deprivation increased ASCT2 levels and decreased SPOP levels, glucose deprivation has minimal, if any, effects on the levels of SPOP and ASCT2. These newly generated data are now presented in Supplementary Fig. 5b, c vs. h, i), and described in the text (page 19).

5) What is the effect of MLN4924 on CK1 δ , does it exert any regulation?

Per reviewer's suggestion, we performed this experiment. The results (Fig. R2) showed that neddylation inhibitor MLN4924 causes a dose-dependent moderate accumulation of CK1 δ , suggesting that CK1 δ could be a substrate of CRLs, an interest subject for future investigation. Given this observation is not the focus of this study, we decided not to include it in this manuscript.

Fig. R2 MDA-MB-231(a) and BT549 (b) cells were treated with MLN4924 at indicated concentrations for 24 h, and then subjected to Western blotting using indicated antibodies.

6) What is the in vivo correlation between SPOP and ASCT2, and the levels of the enzymes involved in neddylation?

We value reviewer's comment. In the data presented in Fig. 7 and Supplementary Fig. 9, we clearly showed an inverse correlation between SPOP and ASCT2 in breast cancer tissues. While the investigation into the correlation between SPOP/ASCT2 and neddylation enzymes (including with one heterodimer E1, two E2s and over 10 E3s) is of interest, we feel that any correlation found will be rather indirect, given that ectopic expression of NEDD8, UBE2M and UBE2F had no effect on the levels of SPOP and ASCT2 (Supplemental Fig. 5 j, k). We believe this is out of scope of current study.

7) What happens with the levels of NEDD8 and the enzymes involved in the neddylation cycle with the levels of glutamine?

Excellent point. We have performed this experiment. The results showed that glutamine deprivation had minimal, if any, effect on the levels of NEDD8, E1 (heterodimer) and two E2s (UBE2M and UBE2F). The newly generated data is shown in Supplementary Fig. 5h, i, and described in the text (pages 10-11).

8) Could SPOP be a neddylation target?

The literature search did not yield any results on SPOP neddylation. To directly address this question, we co-transfected His-tagged NEDD8 and FLAG-tagged SPOP into

HEK293 cells, followed by Ni-NAT beads pull-down of all His-tagged/NEDD8-conjugated proteins, which is then subjected to western blotting. No neddylated SPOP was detected in Ni-NTA pull-down precipitation. We therefore concluded that SPOP is not a neddylation substrate. This newly generated data is included in Supplementary Fig. 6f and described in the text (page 11).

9) Do glutamine levels regulate SPOP neddylation?

While this is an interesting question, we were unable to address it, since SPOP is not a neddylation substrate (see above).

Finally, I would like to take the opportunity to thank all three thoughtful reviewers for their professional review of our manuscript, which made this revision strengthened significantly.

References:

- Bhutia, Y.D., and Ganapathy, V. (2016). Glutamine transporters in mammalian cells and their functions in physiology and cancer. *Biochim Biophys Acta* 1863, 2531-2539.
- Broer, A., Fairweather, S., and Broer, S. (2018). Disruption of Amino Acid Homeostasis by Novel ASCT2 Inhibitors Involves Multiple Targets. *Front Pharmacol* 9, 785.
- Console, L., Scalise, M., Tarmakova, Z., Coe, I.R., and Indiveri, C. (2015). N-linked glycosylation of human SLC1A5 (ASCT2) transporter is critical for trafficking to membrane. *Biochim Biophys Acta* 1853, 1636-1645.
- Im, Y.N., Lee, Y.D., Park, J.S., Kim, H.K., Im, S.Y., Song, H.R., Lee, H.K., and Han, M.K. (2018). GPCR Kinase (GRK)-2 Is a Key Negative Regulator of Itch: L-Glutamine Attenuates Itch via a Rapid Induction of GRK2 in an ERK-Dependent Way. *J Invest Dermatol* 138, 1834-1842.
- Jiang, H., Zhang, N., Tang, T., Feng, F., Sun, H., and Qu, W. (2020). Target the human Alanine/Serine/Cysteine Transporter 2(ASCT2): Achievement and Future for Novel Cancer Therapy. *Pharmacol Res* 158, 104844.
- Scalise, M., Pochini, L., Galluccio, M., Console, L., and Indiveri, C. (2017). Glutamine Transport and Mitochondrial Metabolism in Cancer Cell Growth. *Frontiers in oncology* 7, 306.
- Schulte, M.L., Fu, A., Zhao, P., Li, J., Geng, L., Smith, S.T., Kondo, J., Coffey, R.J., Johnson, M.O., Rathmell, J.C., *et al.* (2018). Pharmacological blockade of ASCT2-dependent glutamine transport leads to antitumor efficacy in preclinical models. *Nature medicine* 24, 194-202.
- Wahi, K., and Holst, J. (2019). ASCT2: a potential cancer drug target. *Expert Opin Ther Targets* 23, 555-558.
- Wang, Q., Hardie, R.A., Hoy, A.J., van Geldermalsen, M., Gao, D., Fazli, L., Sadowski, M.C., Balaban, S., Schreuder, M., Nagarajah, R., *et al.* (2015). Targeting ASCT2-mediated glutamine uptake blocks prostate cancer growth and tumour development. *The Journal of pathology* 236, 278-289.
- Wang, X., Jin, J., Wan, F., Zhao, L., Chu, H., Chen, C., Liao, G., Liu, J., Yu, Y., Teng, H., *et al.* (2019). AMPK Promotes SPOP-Mediated NANOG Degradation to Regulate Prostate Cancer Cell Stemness. *Developmental cell* 48, 345-360 e347.
- Zhang, Z., Liu, R., Shuai, Y., Huang, Y., Jin, R., Wang, X., and Luo, J. (2020). ASCT2 (SLC1A5)-dependent glutamine uptake is involved in the progression of head and neck squamous cell carcinoma. *Br J Cancer* 122, 82-93.

- Zhou, Q., Li, H., Li, Y., Tan, M., Fan, S., Cao, C., Meng, F., Zhu, L., Zhao, L., Guan, M.X., *et al.* (2019). Inhibiting neddylation modification alters mitochondrial morphology and reprograms energy metabolism in cancer cells. *JCI insight* 2019 Feb 21;4(4):e121582. doi: 10.1172/jci.insight.121582
- Zhuang, M., Calabrese, M.F., Liu, J., Waddell, M.B., Nourse, A., Hammel, M., Miller, D.J., Walden, H., Duda, D.M., Seyedin, S.N., *et al.* (2009). Structures of SPOP-substrate complexes: insights into molecular architectures of BTB-Cul3 ubiquitin ligases. *Molecular cell* 36, 39-50.

REVIEWER COMMENTS

Reviewer #1 (Remarks to the Author):

The authors have fully addressed my concerns and I support the publication of the revised manuscript in Nature Communications.

Reviewer #2 (Remarks to the Author):

The reviewers have adequately addressed my concerns.

Reviewer #3 (Remarks to the Author):

The authors respond to all the issues raised for the reviewer

They do a great job